# Novel adverse outcome pathways revealed by chemical genetics in a developing marine fish

Elin Sørhus[1,2]*, John P Incardona[3], Tomasz Furmanek[1], Giles W Goetz[3], Nathaniel L Scholz[3], Sonnich Meier[1], Rolf B Edvardsen[1†], Sissel Jentoft[2,4†]

[1]Institute of Marine Research, Bergen, Norway; [2]Centre for Ecological and Evolutionary Synthesis, University of Oslo, Oslo, Norway; [3]Environmental and Fisheries Science Division, Northwest Fisheries Science Center, National Marine Fisheries Service, Seattle, United States; [4]Department of Natural Sciences, University of Agder, Kristiansand, Norway

**Abstract** Crude oil spills are a worldwide ocean conservation threat. Fish are particularly vulnerable to the oiling of spawning habitats, and crude oil causes severe abnormalities in embryos and larvae. However, the underlying mechanisms for these developmental defects are not well understood. Here, we explore the transcriptional basis for four discrete crude oil injury phenotypes in the early life stages of the commercially important Atlantic haddock (*Melanogrammus aeglefinus*). These include defects in (1) cardiac form and function, (2) craniofacial development, (3) ionoregulation and fluid balance, and (4) cholesterol synthesis and homeostasis. Our findings suggest a key role for intracellular calcium cycling and excitation-transcription coupling in the dysregulation of heart and jaw morphogenesis. Moreover, the disruption of ionoregulatory pathways sheds new light on buoyancy control in marine fish embryos. Overall, our chemical-genetic approach identifies initiating events for distinct adverse outcome pathways and novel roles for individual genes in fundamental developmental processes.

**\*For correspondence:** elin. sorhus@imr.no

[†]These authors contributed equally to this work

**Competing interests:** The authors declare that no competing interests exist.

## Introduction

Catastrophic oil spills, rising water temperatures, ocean acidification, and other large-scale anthropogenic forcing pressures impact the health and survival of myriad marine species in ways that are often unknown. Mechanistic relationships between environmental stress and adverse health outcomes are most readily studied in model laboratory organisms. However, domesticated experimental models may be relatively insensitive to real-world environmental change. Emerging genomic technologies, including high-throughput RNA sequencing (RNA-seq), are providing new opportunities to profile physiological stress in wild, non-model species at a transcriptional level. This approach is premised on the full anchoring of gene expression to physiological or morphological injury phenotypes.

The early life stages of marine fish are particularly vulnerable to pollution and other stressors. However, developmental analyses in wild species can be challenging due to limited access to embryos and larvae. Also limited are molecular and cellular tools for imaging-specific structures (e.g. fluorescent protein-expressing transgenes). This includes, for example, a lack of species-specific probes for visualizing gene expression via *in situ* hybridization. On the other hand, one of the major vertebrate models for studying developmental genetics is a fish. Zebrafish have been a focus for high-throughput experimental techniques for more than three decades. This has yielded a wealth of information on embryonic gene expression patterns, with comparable data often available in chick

**eLife digest** Accidental oil spills are a worldwide threat to ocean life. Fish eggs and larvae are especially vulnerable; therefore oil spills in areas where fish spawn are of great concern. Fish embryos exposed to crude oil grow slower than normal as larvae and juveniles and often show defects in the heart, face and jaw. However, the underlying mechanisms behind these defects are largely unknown.

Working with the Atlantic haddock (*Melanogrammus aeglefinus*), Sørhus et al. have now examined fish embryos and larvae that had been exposed to crude oil, and identified those genes that were more active or less active than normal. The findings add further support to the idea that exposure to crude oil causes heart and face defects because it interferes with how the cells that develop into these structures use calcium ions. Signals sent via calcium ions are not only important for the contraction of muscle cells, but they are also essential for regulation of some genes. So, by interfering with the circulation of calcium ions, crude oil can have consequences for both how muscles work and how genes are regulated.

Sørhus et al. also report two previously uncharacterized defects. Firstly, genes that help to regulate the ion and water content of the tissues were highly affected in young fish exposed to crude oil. Some of the genes were more active than normal, while others were less active. This finding in particular would explain why oil-exposed embryos often accumulate fluids, and suggests that the larvae may have altered buoyancy too. Secondly, the oil-exposed embryos showed signs of a shortage of cholesterol and other fatty molecules. This is most likely because they absorbed less material from their yolk, which could also explain why larvae exposed to crude oil grow more slowly than normal.

Finally, in the future, these newly identified genes connected to crude oil toxicity could be used as diagnostic markers to confirm oil-induced injury in fish, and monitor the health of fish populations in the ocean.

and mouse embryos. The zebrafish platform therefore provides a powerful mechanistic context for anticipating environmental health impacts in marine fish spawning habitats.

Here, we use an RNA-seq approach to assess the effects of crude oil on the early life stages of a cold-water marine species, Atlantic haddock (*Melanogrammus aeglefinus*). Crude oil spills such as the 1989 *Exxon Valdez* (Prince William Sound), 2002 *Prestige* (Spain), and 2010 *Deepwater Horizon* (Gulf of Mexico) continue to threaten fisheries worldwide. Haddock are commercially valuable in Norway and other North Atlantic countries, and they spawn in areas proposed for future crude oil production (e.g. the Lofoten archipelago in Nordland). Similar to many other fish species, haddock embryos are vulnerable to developmental defects from crude oil exposure (Norwegian Sea crude; [*Sørhus et al., 2016b*]). Moreover, we recently showed that RNA-seq applied to normally developing haddock clearly anchored organogenesis phenotypes to the expression of genes involved in determination and differentiation (*Sørhus et al., 2016a*).

Crude oils are complex chemical mixtures, and fish early life stages are especially vulnerable to component polycyclic aromatic hydrocarbons (PAHs) and their alkylated homologues (*Carls and Meador, 2009*; *Adams et al., 2014*). Crude oil-derived PAHs containing three rings disrupt the normal form and function of the embryonic heart, and circulatory failure causes a range of secondary defects (*Incardona et al., 2004*, *2005*). For individual heart muscle cells, the cardiotoxic mechanism involves a blockade of the repolarizing potassium efflux and a reduction in intracellular calcium cycling (*Brette et al., 2014*). The consequent disruption of excitation-contraction (E-C) coupling leads to rhythm and contractility defects at the scale of the developing heart (*Incardona et al., 2009*, *2014*; *Sørhus et al., 2016b*). Mechanisms underpinning morphological defects in other embryonic tissues are still poorly understood.

Based on conventional measures of cardiac function and embryolarval anatomy, zebrafish and haddock respond to Norwegian Sea crude oil in ways that are similar and dissimilar. Both species show characteristic abnormalities including bradycardia, reduced chamber contractility, and fluid accumulation in the vicinity of the heart (edema). This suggests an understanding of zebrafish

developmental genetics will inform the interpretation of changing messenger RNA (mRNA) levels in crude oil-exposed haddock as determined by RNA-seq. This is particularly true for tissue-specific patterns of gene expression that are highly conserved across vertebrates—for example, genes involved in cardiac organogenesis.

Relative to zebrafish, however, haddock are sensitive to much lower concentrations of crude oil and also display a distinct suite of craniofacial defects that cannot be attributed to circulatory failure (*Sørhus et al., 2016b*). There are several reasons to expect divergent effects of crude oil on marine fish embryos and larvae. These are attributable to differences in physiology and life history. For example, accumulation of cardiac edema is a canonical form of crude oil toxicity in both freshwater and marine species. Yet marine embryos are hyposmotic to the surrounding water and hence expected to lose water with circulatory failure. This suggests that PAHs may have distinct impacts on ionoregulation and related processes. Also, unlike zebrafish, many pelagic marine embryos are buoyant and have a characteristic morphology not found in species with demersal (sinking) eggs and larvae. Shelbourne first described a relationship between this unique morphology of pelagic fish larvae, osmoregulation and buoyancy control in the mid-twentieth century (*Shelbourne, 1955*, *1956*, *1957*), but there has been little progress in the decades since, particularly at a molecular scale.

Understanding cause-effect relationships between exposure to environmental contaminants like crude oil and adverse impacts on organismal health are critical for the construction of adverse outcome pathways (AOPs). The development and application of AOPs is an ongoing movement to improve risk assessments. AOPs are derived from detailed toxicological cause-and-effect relationships that span multiple levels of biological organization, ideally from molecular initiating events to species, community or ecosystem scale responses of regulatory concern (e.g. reduction in a fisheries abundance target). AOPs are widely used in risk assessments for both human and environmental (ecological) health (*Ankley et al., 2010*; *Kramer et al., 2011*; *Villeneuve et al., 2014*; *Garcia-Reyero, 2015*) Our long-term aim is to develop AOPs specific to oil spills and fish populations, premised on well-studied early life stage toxicity. AOPs based on crude oil cardiotoxicity in developing fish are already fairly well constructed (*Incardona and Scholz, 2016*) but currently lack details at the molecular level at several steps, particularly in relation to cardiac dysmorphogenesis. We anticipate that identification of changes in gene expression associated with oil-induced developmental defects will further complete these AOPs and expand the molecular toolkit for quantifying oil spill impacts.

In the present study, we used visible developmental abnormalities as phenotypic anchors for evaluating changes in haddock gene expression. We sequenced the full haddock transcriptome at several time points during and after embryonic and larval crude oil exposures. This approach allowed us to explore gene regulation in association with three distinct phenotypes: (1) heart form and function defects, (2) craniofacial deformities, and (3) fluid balance abnormalities and the characteristic pelagic larval form. We also identify perturbations in cholesterol homeostasis linked to poor absorption of yolk as a novel form of crude oil toxicity in marine fish embryos. Our findings are interpreted in the context of highly conserved gene regulation in zebrafish and other vertebrates.

## Results

### Structure of pelagic larvae and visible phenotypes associated with crude oil exposure

At a rearing temperature of 7°C, haddock embryos began hatching at 12 days post-fertilization (dpf). Unlike zebrafish that complete gastrulation (epiboly) before segmentation (somitogenesis) begins, haddock embryos begin forming anterior somites at about 50% epiboly (3 dpf). They subsequently reach the tailbud stage by 6 dpf (~30 somites), have a regular heartbeat by 8 dpf and completion of organogenesis at 10 dpf (*Fridgeirsson, 1978*; *Hall et al., 2004*; *Sørhus et al., 2016a*). Haddock yolk sac larvae have the characteristic morphology associated with ichthyoplankton from pelagic marine habitats (*Figure 1A*), marked by a large marginal finfold that surrounds the larva nearly completely on both the dorsal and ventral sides. The outer epidermis is thus separated from the brain, main body axis muscles, and internal organs by a voluminous subdermal space. This space is filled with extracellular matrix (*Morrison, 1993*) and is continuous with an avascular yolk sac sinus, with connections between the dorsal space and the ventral yolk sac in the vicinity of the pectoral fin

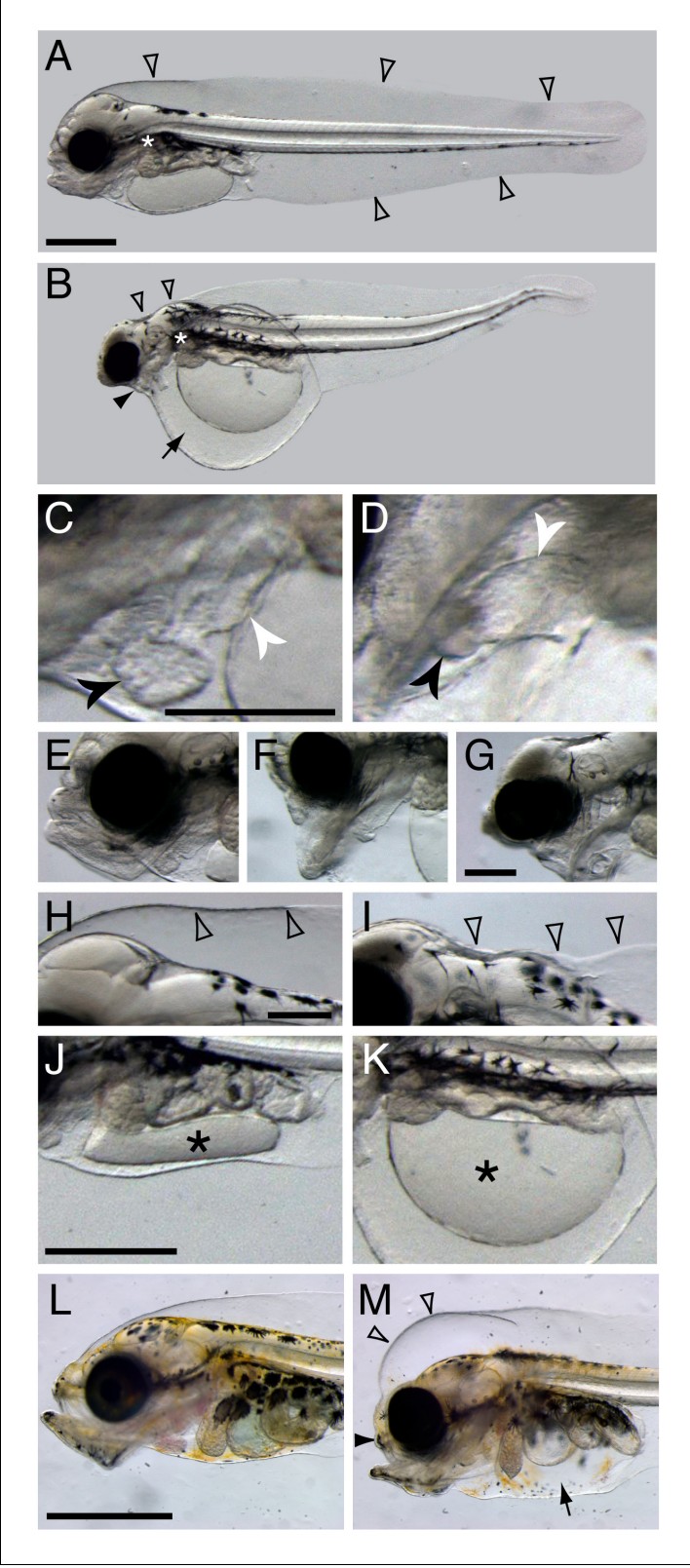

**Figure 1.** Terminal phenotypes after high dose exposure. Control (**A**) and exposed (**B**) three days post hatch (dph) larvae (6 days post embryonic exposure). Open arrowheads in (**A**) indicate the marginal finfold surrounding the larvae and the white asterisk indicate the location of the connection between the dorsal space and the ventral yolk sac in the vicinity of the pectoral fin. In (**B**) the black arrowhead indicates severely reduced craniofacial outgrowth,

*Figure 1 continued on next page*

*Figure 1 continued*

while the black arrow indicates yolk sac edema. The ventricle and atrium in control (C) and embryonically exposed (D) animals are indicated by black and white arrows, respectively. (E) Normal craniofacial structure in control, and (F) moderate and (G) severe craniofacial defects in exposed animals. (H) Normal marginal finfold in control, (I) exposed animals with severe reduction of anterior marginal finfold (open arrowheads). Yolk mass (*) in control (J) and embryonically exposed larvae (K). (L) Control and (M) exposed 18 dph larvae. Open arrowheads indicate increased anterior marginal finfold, black arrowhead indicates reduced upper jaw outgrowth, and black arrow indicates edema formation in the peritoneal cavity in oil-exposed larvae (M). Scale bar: 0.2 mm (C,D; E–G; H–K) and 1 mm (A,B and L,M).

The following figure supplement is available for figure 1:

**Figure supplement 1.** Normal development of liver and lateral line in the severe phenotypes.

(*Shelbourne, 1955*). The subdermal space acts as a reservoir for water balance in order to maintain larval buoyancy (*Shelbourne, 1955*, *1956*, *1957*), with specialized cells regulating ion and water balance (ionocytes or mitochondria rich cells, MRCs) distributed throughout the adjacent epidermis (*Shelbourne, 1957*; *Hirose et al., 2003*; *Hiroi et al., 2005*).

Haddock embryos were continuously exposed to crude oil at environmentally relevant total PAH concentrations of 6.7 ± 0.2 µg/L (high dose) and 0.58 ± 0.05 µg/L (low dose), and intermittently at 6.1 µg/L (pulse dose). This yielded internal total PAH doses of 3.0 ± 1.3 µg/g wet weight, 0.19 ± 0.02 µg/g, and 0.22 ± 0.06 µg/g, respectively (see [*Sørhus et al., 2016b*] for experimental details). Although the low and pulsed exposures led to similar total PAH accumulation in embryos, phenotypes were slightly more severe in the latter due to relatively higher exposure concentrations during critical windows of early development (*Sørhus et al., 2016b*). The embryonic exposure began at 2 dpf and ended shortly after the end of organogenesis at 10 dpf, just prior to hatch. Larvae were exposed to the same regimen from day of hatch to 18 days post hatch (dph). As expected, tissue PAH accumulation was lower than for embryos, with 0.81 ± 0.18 µg/g, 0.086 ± 0.015 µg/g, and 0.081 ± 0.024 µg/g, for high, low, and pulse doses, respectively. Except where indicated below, phenotypes were quantified as described in detail previously (*Sørhus et al., 2016b*), with 96% of high-dose animals showing abnormal phenotypes, ranging to ~60% for pulse dose and ~35% for the low dose. Representative terminal phenotypes in high dose are shown in *Figure 1* for the embryonic exposure at 3 dph and larval exposure at 18 dph. Grossly, as in other species, crude oil exposure led to defects in cardiac function and morphology and accumulation of edema around the heart and in the yolk sac (*Figure 1B*). Defects in cardiac morphology included a failure to properly loop the atrial and ventricular chambers from a linear to an adjacent orientation, and reduced size of the ventricle (*Figure 1C,D*). In addition, oil-exposed haddock embryos displayed craniofacial defects in their larval stages that ranged in severity (*Figure 1E–G*), from marked reductions in upper jaw/skull base structures (*Figure 1F*) to near complete lack of upper and lower jaws (*Figure 1G*). Moreover, the anterior portion of the dorsal marginal finfold was collapsed or missing and the hindbrain ventricle typically failed to fill with cerebrospinal fluid in embryonically exposed larvae with severe edema (*Figure 1H, I*). Finally, in more severely affected embryos, a failure of yolk absorption was obvious at 3 dph (*Figure 1J,K*). Even in mildly affected embryos, yolk absorption was reduced after hatch as assessed by measuring the two-dimensional area of the yolk mass in lateral images (control yolk area control 0.63 ± 0.06 mm$^2$, low-dose group 0.90 ± 0.11, high-dose group 1.2 ± 0.3; mean ± s.d., ANOVA p<0.0001). In contrast, there was no measurable difference in yolk area at day of hatch. After larval exposure, the primary morphological defects were reduced jaw growth and edema accumulation (*Figure 1L,M*), the latter in the peritoneal cavity. In contrast to embryos, the dorsal anterior subdermal space accumulated fluid in larvae and did not collapse.

Abnormal phenotypes relating to the formation of edema, heart function and morphogenesis, craniofacial structure, and yolk absorption manifested at different developmental time points during embryonic exposure and afterwards when embryos were transferred to clean water for hatching (*Figure 2*). Samples were collected for transcriptome profiling at four embryonic stages (E1-4, *Figure 3A*) and at two stages post-hatch (E5-6, *Figure 3A*). At 6 dpf (E2 sampling point; ~30 somites), embryos exposed to the high dose were indistinguishable from controls (*Figure 2A*). By 8

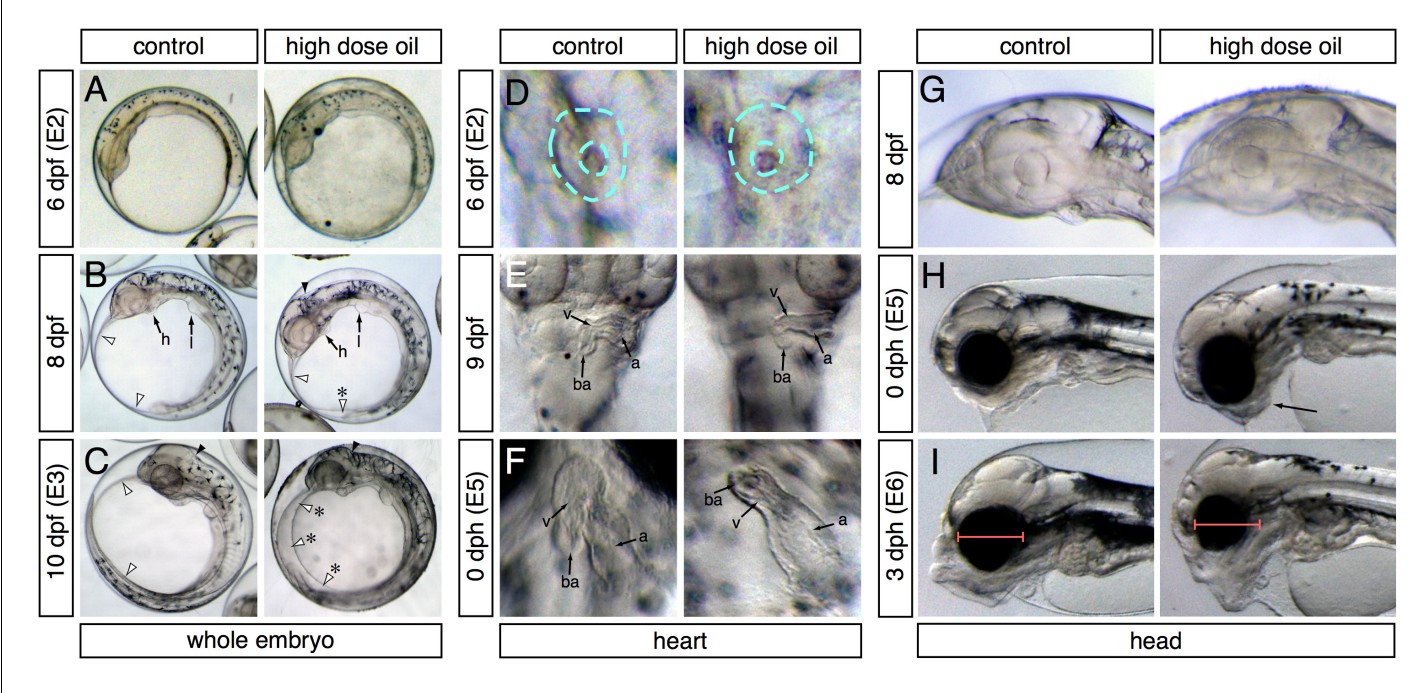

**Figure 2.** Appearance of phenotypes over time. In each panel control and high-dose-exposed embryos are shown on the left and right, respectively. (A–C) Lateral overview of whole embryos showing accumulation of edema (anterior to the left). (A) 6 dpf/E2 sampling point. (B) 8 dpf (between E2 and E3 sampling points). Heart (h) and liver bud (l) are indicated. White arrowheads indicate outer margins of the yolk sac membranes; asterisk indicates small pocket of edema. Black arrowheads indicate the hindbrain ventricle. (C) 10 dpf/E3 sampling point. Arrowheads same as (B); asterisks indicate expanded yolk sac edema. (D–E) High-magnification ventral views of the heart (anterior at top). (D) 6 dpf/E2. Dashed turquoise lines indicate outer border and lumen of midline cardiac cone. (E) 9 dpf (between E2 and E3). Arrows indicate the atrium (a), ventricle (v) and bulbus arteriosus (ba). (F) 0 dph (E5 sampling point). Chambers indicated as in (E). (G–I) Lateral views of the developing head (anterior to the left). (G) 8 dpf (between E2 and E3). (H) 0 dph (E5). Arrow indicates abnormal lower jaw cartilages in oil-exposed larva. (I) 3 dph (E6 sampling point). Red bars indicate difference in eye diameter between control and exposed larvae.

dpf, small accumulations of edema could be observed in the yolk sac of oil-exposed embryos, but their hindbrain ventricles were 'inflated' with fluid (*Figure 2B*). By 10 dpf (E3 sampling point), edema was evident in most embryos, typically filling the space above the yolk between the anterior of the head and the tail, and hindbrain ventricles lacked fluid (*Figure 2C*). Similarly, at 6 dpf/E2, heart development appeared unaffected in oil-exposed embryos and was at the un-rotated midline cone stage (*Figure 2D*). By 9 dpf (one day before sample E3), hearts in both control and high-dose-exposed embryos had rotated and were beginning to loop, but ventricular walls already appeared slightly thinner (*Figure 2E*) and heart rate was significantly slower (20 + 6 beats/min compared to 26 ± 3 beats/min for controls; [*Sørhus et al., 2016b*]). By day of hatch (E5 sampling), a high percentage (54%) of exposed embryos showed un-looped hearts with small, silent ventricles (*Figure 2F*).

Onset of craniofacial abnormalities took a longer course. At 8 dpf (2 days before sample E3), head structures appeared identical in control and exposed embryos (*Figure 2G*). At hatch (E5 sample), jaw structures appeared somewhat abnormal (*Figure 2H*) but became much more strikingly severe by 3 dph (E6 sample; *Figure 2I*). Notably, the eyes appeared smaller in exposed embryos by 3 dph (*Figure 2I*). We did not quantify this effect, because it was demonstrated earlier in zebrafish that small eyes result from loss of cardiac function by either genetic or chemical means (*Incardona et al., 2004*), and hence, this phenotype is not specific to crude oil toxicity.

Other organs and structures were apparently unaffected by oil exposure. For example, the development of the liver and lateral line neuromasts progressed normally even in the most severely impacted larvae that were exposed as embryos (*Figure 1—figure supplement 1A–D*).

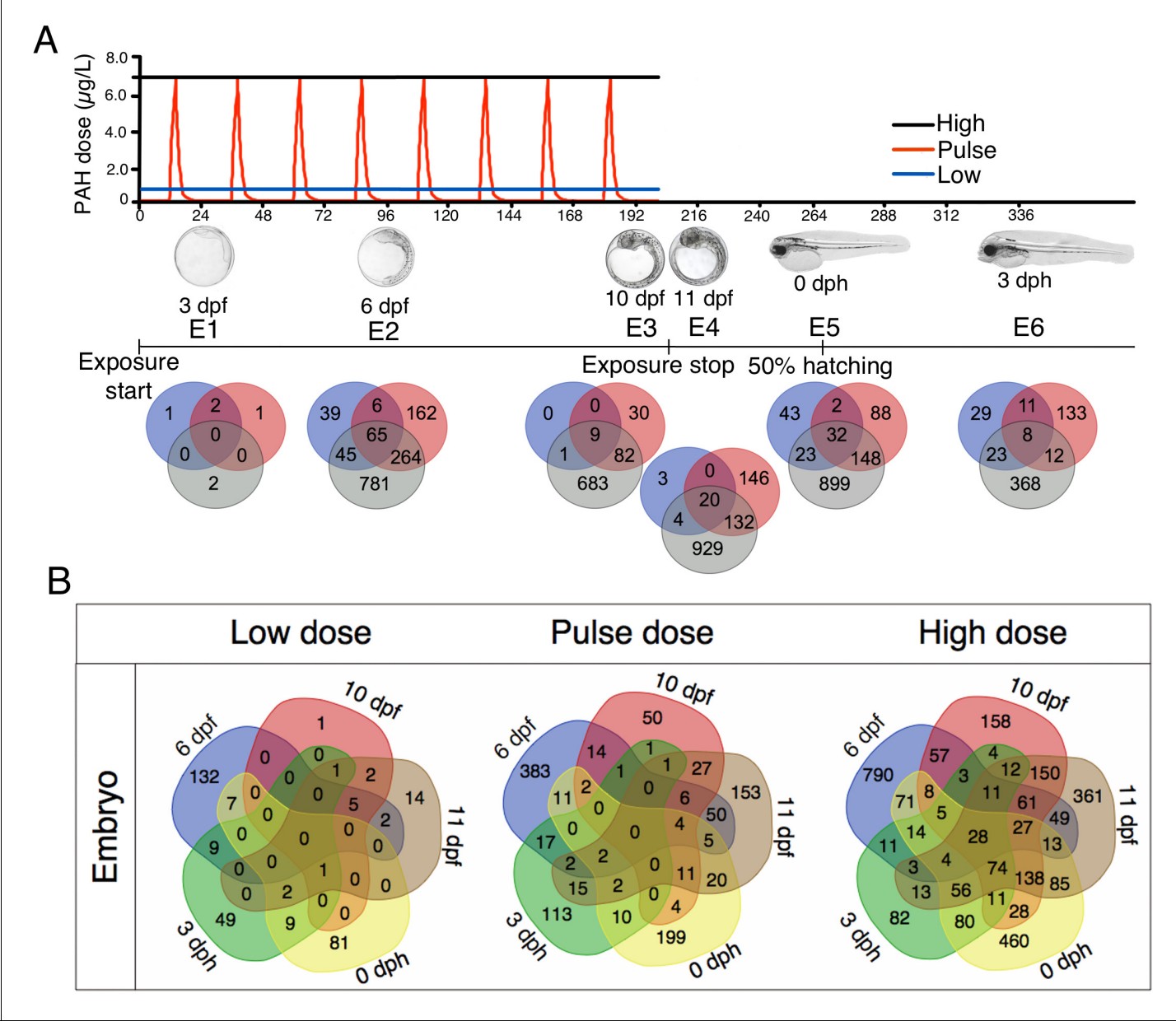

**Figure 3.** Exposure regimes and differentially expressed genes (DEGs) during embryonic development. (**A**) Embryos were exposed to a continuous high dose (black line; 6.7 ± 0.2 µg/L TPAH), a pulsed dose (red line; 0.09 ± 0.02–6.8 ± 1.0 µg/L TPAH) and a continuous low dose (blue line; 0.58 ± 0.05 µg/L TPAH) of crude oil. Photos indicate normal developmental progress at each of six sampling time points (E1–E6). Venn diagrams show shared and exclusive DEGs for each of the three oil exposures at E1–E6. (**B**) Venn diagrams illustrating the number of shared and exclusive DEGs at each stage in development up to hatching for the three exposure regimes.

The following figure supplements are available for figure 3:

**Figure supplement 1.** Most regulated KEGG pathways.

**Figure supplement 2.** Comparison of mRNA read count data with real-time qPCR for selected genes during and after embryonic exposure.

# Oil-induced changes in gene expression during embryonic development

Relative to unexposed controls, differently expressed genes (DEGs) in oil-exposed fish were defined as having significantly ($p < 0.05$) higher or lower levels of expression. The number of exclusive and shared DEGs varied across exposure regime and haddock developmental age (*Figure 3B*). After 24 hr of oil exposure (sampling stage E1; 3 dpf), relatively few genes were differentially expressed, and most were significantly downregulated (Supporting dataset 1, *Sørhus et al., 2017*). From sampling point E2 (6 dpf) through E6 (three days post hatch, dph), however, the number of DEGs was substantial, particularly in response to the high dose exposure. With the exception of the initial sampling point (E1), a total of 28 DEGs were shared across all embryonic stages (*Table 1*). As expected, the largest category (nine DEGs) included genes associated with stress response and xenobiotic metabolism. The remaining genes play a role in tyrosine catabolism, myofibrillar establishment and cardiac tissue repair, central nervous system (CNS) function and degeneration, ATP metabolism, and cholesterol synthesis.

**Table 1.** Genes expressed at all stages during and after embryonic exposure (E2–E6) in high dose group. SP, swissprot; GB, genebank; IE; increased expression; DE; decreased expression.

| Cod ID | Swissprot annotation | SP ID | GB ID | Category | Regulation |
|---|---|---|---|---|---|
| ENSGMOG00000018302 | Fumarylacetoacetase | faaa | fah | Tyrosine metabolism | IE |
| ENSGMOG00000000318 | Cytochrome P450 1A1 | cp1a1 | cyp1a1 | xenobiotic metabolism and stress | IE |
| ENSGMOG00000012518 | Glutathione S-transferase P | gstp1 | gstp1 | xenobiotic metabolism and stress | IE |
| ENSGMOG00000016016 | Glutathione S-transferase omega-1 | gsto1 | gsto1 | xenobiotic metabolism and stress | IE |
| ENSGMOG00000018752 | 3-hydroxyanthranilate 3,4-dioxygenase | 3hao | haao | xenobiotic metabolism and stress | IE |
| ENSGMOG00000006796 | 3-beta-hydroxysteroid-Delta(8),Delta(7)-isomerase | ebp | ebp | xenobiotic metabolism and stress | IE |
| ENSGMOG00000007636 | Glutamine synthetase | glna | glul | xenobiotic metabolism and stress | IE |
| ENSGMOG00000015234 | Heat shock protein HSP 90-alpha | h90a1 | hsp90a.1 | xenobiotic metabolism and stress | IE |
| ENSGMOG00000012029 | Peptidyl-prolyl cis-trans isomerase | ppia | - | xenobiotic metabolism and stress | IE |
| ENSGMOG00000000218 | Ammonium transporter Rh type A OS=Mus | rhag | rhag | xenobiotic metabolism and stress | Mainly IE |
| ENSGMOG00000003353 | Ferritin, middle subunit | frim | - | xenobiotic metabolism and stress | IE |
| ENSGMOG00000018206 | Filamin-C | flnc | Flnc | myofibrillar establishment and repair | IE |
| ENSGMOG00000001317 | Iron-sulfur cluster assembly enzyme ISCU, mitochondrial | iscu | Iscu | cardiac defects | IE |
| ENSGMOG00000010446 | Fatty acid-binding protein, heart | fabph | fabp3 | cardiac defects and repair | IE |
| ENSGMOG00000007115 | Lanosterol 14-alpha demethylase | cp51a | cyp51a1 | Cholesterol syntheis | IE |
| ENSGMOG00000005565 | Squalene monooxygenase | erg1 | Sqle | Cholesterol syntheis | IE |
| ENSGMOG00000018991 | Farnesyl pyrophosphate synthase | fpps | fdps | Cholesterol syntheis | IE |
| ENSGMOG00000005774 | 3-hydroxy-3-methylglutaryl-coenzyme A reductase | hmdh | hmgcr | Cholesterol syntheis | IE |
| ENSGMOG00000015657 | Epididymal secretory protein | npc2 | npc2 | Cholesterol syntheis | IE |
| ENSGMOG00000001249 | Putative adenosylhomocysteinase | sahh3 | ahcyl2 | cardiac defects | DE |
| ENSGMOG00000013374 | Peptide Y OS=Dicentrarchus | py | - | CNS function and development | IE |
| ENSGMOG00000014820 | Complement C1q-like protein | c1ql2 | c1ql2 | CNS function and development | IE |
| ENSGMOG00000017148 | Augurin-A OS=Danio rerio | augna | zgc:112443 | CNS function and development | IE |
| ENSGMOG00000001072 | C-4 methylsterol oxidase | erg25 | sc4mol | CNS function and development | IE |
| ENSGMOG00000013980 | Fatty acid-binding protein, brain | fabp7 | fabp7 | CNS function and development | IE |
| ENSGMOG00000014938 | Maltase-glucoamylase, intestinal | mga | mgam | ATP metabolism | IE |
| ENSGMOG00000003530 | ADP/ATP translocase | adt3 | slc25a6 | ATP metabolism | DE |
| ENSGMOG00000006172 | IEF0762 protein C6orf58 homolog | cf058 | - | not known | IE |

## Oil-induced changes in gene expression during larval development

Haddock larvae under the same three exposure regimes were transcriptionally profiled at five distinct developmental stages (L1-5; *Figure 4A*). Relative to embryos, transcriptional responses to crude oil-exposed haddock larvae were more subtle at 1, 2, and 9 dph (L1-3). This was followed by marked changes in gene expression at 14 dph (L4) for the high dose and for all treatments at 18 dph (L5) (Supporting dataset 2, *Sørhus et al., 2017*). In the high-dose group, nearly 1000 of the >3000

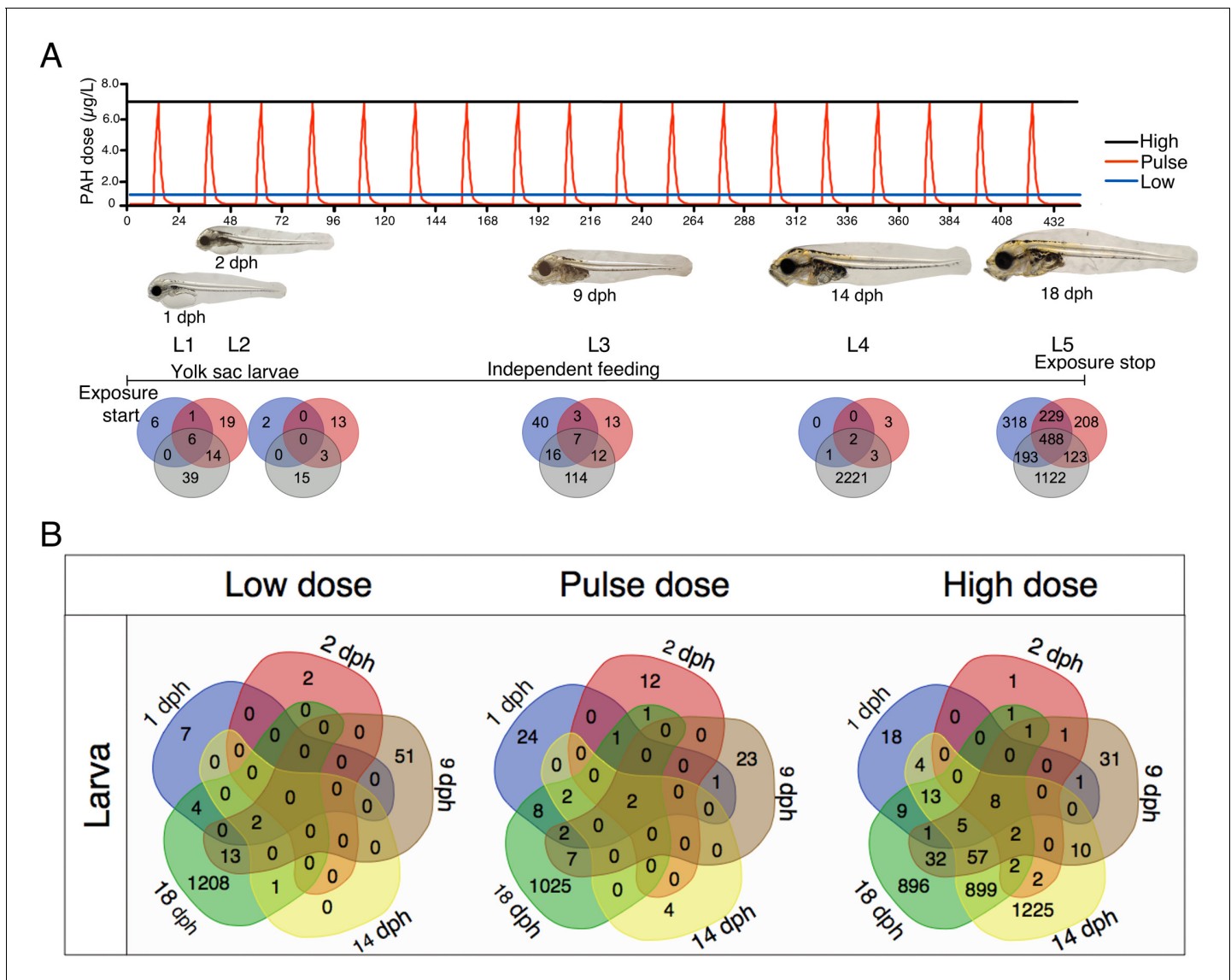

**Figure 4.** Exposure regimes and differentially expressed genes (DEGs) during larval development. (**A**) Larvae were exposed to a continuous high dose (black line; 7.6 ± 0.7 µg/L TPAH), a pulsed dose (red line; 0.3 ± 0.3–6.1 ± 0.5 µg/L TPAH), and a continuous low dose (blue line; 0.65 ± 0.08 µg/L TPAH) of crude oil. Photos indicate normal developmental progress at each of five sampling time points (L1–L5). Venn diagrams show shared and exclusive DEGs for each of the three oil exposures at L1–5. (**B**) Venn diagrams illustrating the number of shared and exclusive DEGs at each larval stage for the three exposure regimes.

The following figure supplements are available for figure 4:

**Figure supplement 1.** Most regulated KEGG pathways.

**Figure supplement 2.** Comparison of mRNA read count data with real time qPCR for selected genes during larval exposure.

DEGs at L4 and L5 were shared (*Figure 4B*). However, for the high-dose treatment, only eight genes were shared across all larval stages. As expected, five of these genes are involved in xenobiotic metabolism (*Table 2*).

## General patterns of gene expression in response to crude oil

Read count data from the RNA-Seq closely matched expected abundances based on tissue-specific expression patterns for orthologous genes in zebrafish, available in the expression database at www.zfin.org (*Supplementary file 1A*), and generally correlated with the overall mass of the contributing tissues. Genes known to have tightly restricted cardiac expression generally had read count values below 100, with *bmp10* and *kcnh2* just above detection limits (10 reads). At the onset of expression in zebrafish, *bmp10* transcripts are detected by *in situ* hybridization in perhaps fewer than 100 cells (*Laux et al., 2013*). In contrast, *bmp4* is more widely expressed in zebrafish at the segmentation stage, including the eye, tailbud, and epidermis in addition to the heart. As expected, this gene had a correspondingly higher read count (265) in haddock. Genes expressed more strongly throughout the entire heart tube had read counts above 60 (e.g. *nkx2.5* at 85 and *fhl2* at 65). The cardiac-specific Serca2 isoform (*atp2a2*) had a read count of 312, while the isoform expressed in skeletal muscle (*atp2a1*) had a read count of 9114. Similarly, read counts for the atrial-specific isoform of myosin heavy chain (*myh6*) and the major skeletal muscle isoform *myh1* were 176 and 2543, respectively. Genes expressed in the neural tube, a tissue mass much larger than the heart but less than the myotomes, had intermediate read counts that also fit with their relative expression patterns. For example, *pax6* and *nkx2.2* had read counts of 1773 and 333, respectively, with *pax6* expressed in a fairly wide dorsal domain of the neural tube, and *nkx2.2* expressed in a narrower ventral region.

We characterized pathways affected by oil exposure using three methods: extensive manual curation, KEGG Pathway Mapping, and Ingenuity Pathway Analysis (IPA; see Materials and methods). As detailed in the following sections, our manual curation identified specific patterns of gene expression in the context of cardiotoxicity, craniofacial deformities, disrupted ion and water balance, and disrupted cholesterol homeostasis. These same pathways were identified with statistical rigor using IPA and KEGG. Moreover, IPA demonstrated enrichment for these pathways at developmental time points that preceded the onset of visible phenotypes, and a lack of enrichment for pathways associated with structures that were phenotypically normal (*Table 3*).

At all stages, the IPA subcategory of Organismal Development or Embryonic Development (henceforth combined as Development) was in the top 5 Diseases and Bio Functions category under Physiological System Development and Function with p values ranging from $10^{-3}$ to $10^{-19}$ (*Tables 2* and *3*). Counts of the number of pathways specifically involving the cardiovascular system showed that no pathways were affected at E1 (3 dpf, 50% epiboly/cardiac progenitor stage), while the heart represented 22% of the affected Development pathways at 6 dpf/E2, the cardiac cone stage at which there was no visible phenotype (*Table 3*). The number of cardiovascular pathways fell to 5.7% at 10 dpf/E3, by which point hearts were visibly abnormal, falling to only one or two affected pathways at hatching stages (E5 and E6).

**Table 2.** Genes expressed at all stages during larval exposure (L1–L5) in high-dose group. SP, swissprot; GB, genebank; IE; increased expression; DE; decreased expression.

| Cod ID | Swissprot annotation | SP ID | GB ID | Category | Regulation |
|---|---|---|---|---|---|
| ENSGMOG00000009114 | Aryl hydrocarbon receptor repressor | *ahrr* | *ahrr* | Xenobiotic metabolism | IE |
| ENSGMOG00000020141 | Cytochrome P450 1B1 | *cp1b1* | *cyp1b1* | Xenobiotic metabolism | IE |
| ENSGMOG00000006842 | Cytochrome P450 1B1 | *cp1b1* | *cyp1b1* | Xenobiotic metabolism | IE |
| ENSGMOG00000019790 | Cytochrome P450 1B1 | *cp1b1* | *cyp1b1* | Xenobiotic metabolism | IE |
| ENSGMOG00000000318 | Cytochrome P450 1A1 | *cp1a1* | *cyp1a1* | Xenobiotic metabolism | IE |
| ENSGMOG00000014967 | Keratinocyte growth factor | *fgf7* | *fgf7* | Myocardial development and tissue repair | IE |
| ENSGMOG00000020500 | Forkhead box protein Q1 | *foxq1* | *foxq1* | Transcription factor | IE |
| ENSGMOG00000000218 | Ammonium transporter Rh type A | *rhag* | *rhag* | Gas transport | IE |

**Table 3.** Time course of pathway enrichment relating to affected and unaffected developmental and functional phenotypes.

| Phenotype[†] | Development stage* | | | | | |
|---|---|---|---|---|---|---|
| | 3 dpf/E1 | 6 dpf/E2 | 10 dpf/E3 | 11 dpf/E4 | 0 Dph/E5 | three Dph/E6 |
| Cardiovascular | 0 (0/8) | 22.4 (11/49) | 5.7 (4/70) | 7.0 (4/57) | 4.7 (2/43) | 2.1 (1/48) |
| Craniofacial | 0 (0/8) | 12.2 (6/49) | 10 (7/70) | 5.3 (3/57) | 7.0 (3/43) | 2.1 (1/48) |
| Liver | 12.5 (1/8) | 0 (0/49) | 5.7 (4/70) | 8.8 (5/57) | 0 (0/43) | 0 (0/48) |
| Eye | 0 (0/8) | 4.1 (2/49) | 20 (14/70) | 48.6 (17/35) | 51.2 (22/43) | 50.0 (24/48) |
| Osmoregulation | – | 43.3 (13/30) | 29.3 (12/41) | 15.0 (3/20) | 16 (4/25) | – |
| Cholesterol | 0/30 | 0 (0/27) | 27.1 (13/48) | 31.3 (10/32) | 25.5 (12/47) | – |
| Lipid | 0/30 | 40.7 (11/27) | 35.4 (17/48) | 50.0 (16/32) | 48.9 (23/47) | – |

*Percentage of total enriched pathways (absolute values).

[†]Numbers of affected pathways representing Cardiovascular, Craniofacial, Liver and Eye were extracted from the combined Development category in IPA results; numbers of pathways representing osmoregulation/ion transport were extracted from the Molecular Transport category; numbers of pathways affecting Cholesterol/sterol metabolism and other non-cholesterol lipids (Lipid) were extracted from the Lipid Metabolism category.

Pathways specifically related to head, face, or skull development were similarly enriched at all stages except 3dpf/E3, representing 12% and 10% at 6 dpf/E2 and 10 dpf/E3, prior to visible differences in head structures. Importantly, at 6 dpf/E2, prior to the onset of both visible cardiac and craniofacial defects, the top 10 enriched pathways under Development included three involving head development and two involving heart development (p values $10^{-6}$ to $10^{-12}$; *Supplementary file 1B*). In contrast, pathways relating to liver development were enriched at 5.7% and 8.8% at only two time points, 6dpf/E3 and 10 dpf/E4, and these did not appear in the top 10. Moreover, inspection of individual DEGs associated with those pathways showed genes involved in lipid transport rather than bona fide regulators of liver development (see below). The single pathway relating to the liver at 3 dpf/E1 was represented by a single gene, *cyp1a*. At these stages, these lipid transport genes are most strongly expressed in the yolk syncytial layer. Notably, IPA also detected larger scale enrichment of eye genes, almost all down-regulated, accounting for roughly 50% of developmental pathways during pigmentation of the retina, but prior to obvious differences in eye sizes after hatch.

Genes associated with osmoregulation were identified by IPA under the Molecular Transport category (Diseases and Bio Functions, Molecular and Cellular Functions). We quantified pathways relating to specific ions (e.g., Na$^+$, K$^+$), inorganic ions, and metals (*Table 3*). At E1/3 dpf Molecular Transport was not in the top five affected pathways, but became enriched at 43% at E2/6 dpf, with primarily down-regulation of genes prior to the onset of visible edema. These Molecular Transport pathways remained significantly enriched (29%, 15%, 16%) until onset of hatch (E5). By 3 dph/E6, Molecular Transport pathways dropped below the top 5.

Pathways related to cholesterol and other lipids (e.g. phospholipids, fatty acids) followed a similar pattern as osmoregulation. As for Molecular Transport, Lipid Metabolism was consistently in the top five Molecular and Cellular Functions category. Pathway enrichment was overall at the highest levels for Lipid Metabolism. We separately quantified individual pathways relating to sterols (e.g. cholesterol synthesis, cholesterol transport, steroid biogenesis) and other fundamental (non-signaling) lipids (e.g. fatty acid synthesis and transport, glycerolipids, phospholipids) (*Table 3*). General lipid metabolism pathways were highly enriched at 6 dpf/E2 (41%) and remained high (35–50%) until 3 dph/E6 when Lipid Metabolism pathways fell below the top 5. Cholesterol metabolism pathways were first enriched at 10 dpf/E3 at 27%, remaining at 31% and 26% until 3 dph/E6, when they also fell below the top 5. Notably, all Lipid Metabolism pathways were enriched prior to measureable detection of reduced yolk absorption at 3 dph.

Pathway enrichment was dose-dependent and clearly associated with the frequencies of abnormal phenotypes (*Supplementary file 1C*). For example, the combined general Development categories at 6 dpf included 203, 121, and 0 pathways (among the top five general categories) for the high, pulse, and low doses, respectively. At this stage, Molecular Transport pathways were enriched at levels of 117, 80, and 0 for the high, pulse, and low doses, respectively. At 10 dpf/E3, the Lipid Metabolism category included 115, 28, and 5 pathways for the high, pulse, and low doses, respectively.

Finally, Cardiotoxicity pathways were prominent for nearly all times points for each dose (*Supplementary file 1C*). For high, pulse, and low doses, numbers of enriched pathways at 6 dpf/E2 were, respectively, 63, 33, and 17; at 10 dpf/E3, 79, 10, and 5; at 11 dpf/E4, 92, 40, and 9; at 0 dph/ E5, 69, 46, and 26; and at 3 dph/E6, 34, 19, and 12.

We identified 10 individual genes (*Supplementary file 1D*) and KEGG pathways (*Figure 3—figure supplement 1* and *Figure 4—figure supplement 1*) that were the most highly responsive (highest positive or negative fold change) to the high oil treatment regime relative to unexposed control fish. Briefly, in both embryos and larvae, DEGs involved in xenobiotic metabolism and stress response were highly represented. Genes involved in the development and function of neural networks and cholesterol/steroid biosynthesis were upregulated, while genes involved in intracellular calcium signaling were primarily downregulated.

In order to investigate both pathways with numerous and few genes, we chose two different approaches for KEGG pathways analysis (1) Total: Pathways with the highest number of DEGs ≥ 2 FC (*Figure 3—figure supplement 1A* and *Figure 4—figure supplement 1A*) and (2) Normalised: Pathways with the largest fraction of DEGs ≥2 FC/ Total number of genes in pathway) (*Figure 3— figure supplement 1B* and *Figure 4—figure supplement 1B*).

During embryonic exposure pathways associated with PAH metabolism were represented among the most affected. Indicative of disrupted osmoregulation and ion channel blockade, secretion pathways and calcium signaling showed decreased expression at the earliest stages. Further, we observed increased expression in steroid metabolism and biosynthesis pathways suggesting an effect on cholesterol metabolism (*Figure 3—figure supplement 1B*). Post exposure, we observed increased expression of several pathways suggestive of an inflammatory response (protein digestion and degradation, influenza A, antigen processing and presentation), while expression of genes in phototransduction pathway was decreased (*Figure 3—figure supplement 1A*).

During larval exposure at the first three sampling stages a small number of genes, and thus, pathways were regulated and most were participating in PAH metabolism. Consistent with total number of DEGs (*Figure 4B*), stage L4 and L5 included pathways with higher number of DEGs ≥ 2 FC. Most noticeable was the decreased expression in calcium signaling pathway and hypertrophic cardiomyopathy (HCM) pathway at 14 dph and decreased expression in pancreatic secretion and protein digestion and absorption (*Figure 4—figure supplement 1A*) and steroid biosynthesis pathways (*Figure 4—figure supplement 1B*) at 18 dph.

Finally, four genes stood out as unique for (1) being highly upregulated in both embryos and larvae, (2) their non-affiliation with a larger network or pathway, and (3) their potential connections to visible phenotypes. These included collagen and calcium-binding EGF-like domain 1 (*ccbe1*), the ammonia transporter *rhag*, forkhead box transcription factor *foxq1*, and fibroblast growth factor *fgf7* (*Table 1*, *Supplementary file 1D*). The following sections identify specific patterns of gene expression in the context of cardiotoxicity, craniofacial deformities, disrupted ion and water balance, and disrupted cholesterol homeostasis.

## Genes associated with defects in cardiac function and morphogenesis

As noted above, early formation of the heart was not affected by crude oil exposure, but morphological defects followed after functional defects were first observed at 9 dpf (bradycardia). In addition, morphology became more severely impacted over time, with later defects including failure of looping and poor ventricular growth becoming prominent by 0 dph/E5. There are several possible etiologies for ventricular size reduction. For example, although the precise mechanism by which intracellular calcium regulates embryonic cardiomyocyte proliferation is still unknown, a disruption of calcium cycling could reduce proliferation, thereby yielding fish with smaller hearts (*Rottbauer et al., 2001*; *Ebert et al., 2005*). We therefore focused on genes involved in cardiac morphogenesis. The earliest alteration was a fourfold increase in the signaling molecule, *bmp10* at 6 dpf/E2 while the heart was at the midline cone stage, and appeared unaffected in oil-exposed embryos (*Figure 5*, *Supplementary file 1E*). IPA also identified Bmp signaling as a significantly enriched pathway at this time point, under the Organismal Development category. Elevation of *bmp10* was followed by the upregulation of the cardiac transcription factor *nkx25*, to a threefold increase at 10 dpf/E4 and a nearly sixfold increase at 11dpf/E5, when the heart was beating regularly. At hatch (0 dph), the expression of the transcription factor *tbx3* was elevated eightfold. Lastly, atrial natriuretic factor (*nppa*), a key homeostatic regulator of contractility, was downregulated by

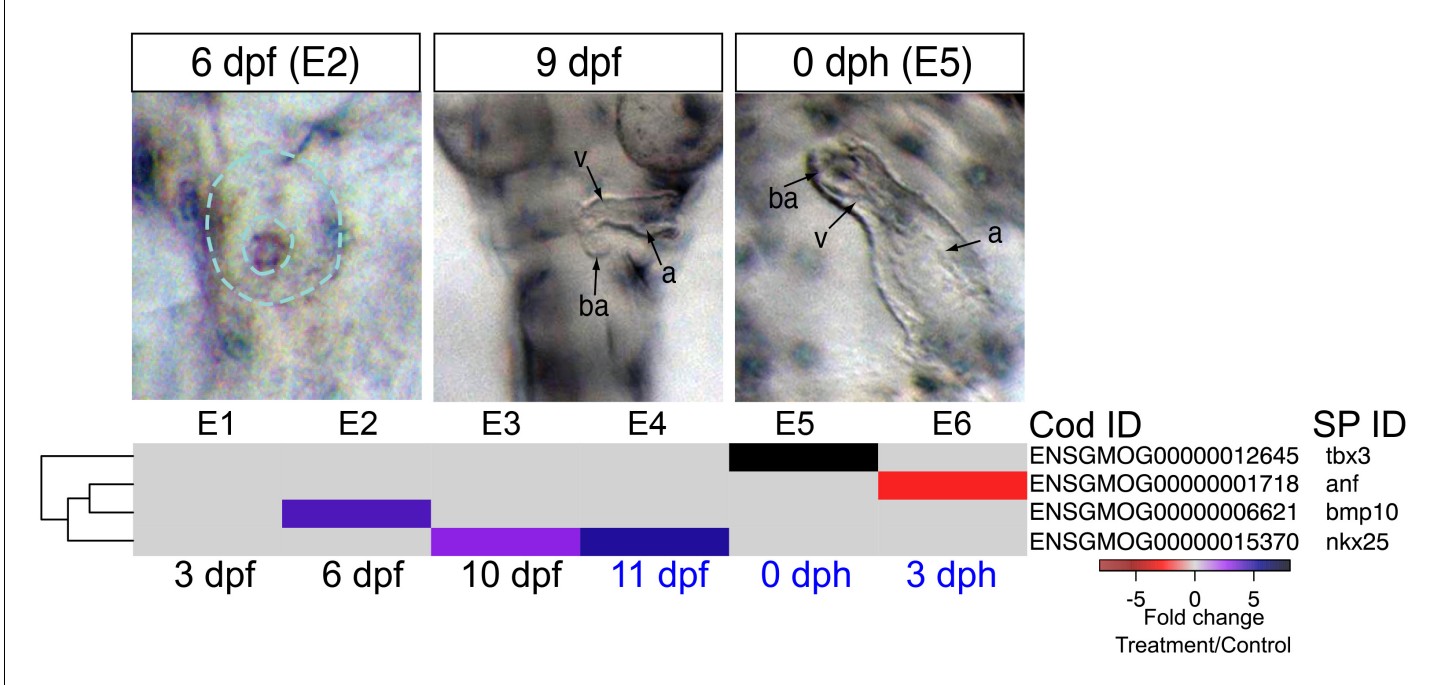

**Figure 5.** DEGs involved in cardiogenesis. Regulation of genes involved in cardiogenesis during and after embryonic exposure. Purple: increased expression, red: decreased expression in exposed group.

2.3-fold in larvae at 3 dph (*Figure 5*, *Supplementary file 1E*). Notably, overexpression of *bmp10*, *nkx25*, or *tbx3* is associated with serious heart defects in other vertebrates (*Chen et al., 2006*; *Ribeiro et al., 2007*; *Tu et al., 2009*).

Crude oil exposures caused functional defects in the developing haddock heart, in the form of bradycardia, ventricular asystole and decreased contractility in embryos and partial atrio-ventricular conduction blockade in larvae (*Sørhus et al., 2016b*). This is consistent with disruption of the rhythmic fluxes of $Ca^{2+}$ and $K^+$ ions that regulate E-C coupling in heart muscle cells (*Brette et al., 2014*). We therefore focused on genes associated with cardiomyocyte membrane potential and intracellular $Ca^{2+}$ cycling—for example, sarcoplasmic reticulum calcium ATPases (SERCAs) and the ryanodine receptor (RyR) ([*Sørhus et al., 2016a*], *Supplementary file 1F* and *1G* ). We found three paralogs for *at2a2* (Serca2) that were present at very different read count values (~300, 900, and 4000 at 6 dpf/E2). The two more abundant paralogs were transiently down-regulated in oil-exposed embryos at 6 dpf/E2, prior to the onset of functional and morphological defects, while the third paralog was down-regulated at 0 dph/E5 (*Figure 6*, *Supplementary file 1H*). Similarly, there were four *nac1* paralogs that were all low abundance, and one was transiently down-regulated with the *at2a2* genes at 6 dpf. Finally, the *kcnh2* gene contributing to the repolarizing K+ current was down-regulated ninefold at hatch/E5. There were effects on a few other E-C coupling genes, but these had very high read counts, and are therefore likely to be associated with skeletal muscle. These included two *at2a1* (serca1) paralogs that had opposite responses, and *atpa*.

A different picture emerged from the larval exposure. Changes in expression of E-C coupling genes occurred after the onset of functional defects (AV block arrhythmia). No changes in cardiac E-C coupling genes were observed at the L3/9 dph time point when larvae showed AV block. Six days later (L4), there was fourfold down-regulation of a *nac1* paralog with the lowest read count value and an *at2a2* paralog with the highest value. At this stage there was up-regulation of two *kcnj2* paralogs (encoding potassium channels), two high abundance *at2a1* paralogs, and a low abundance *scn2a* paralog. At 18 dph/L5, one paralog each of *at2a1* and *kcnj2* remained elevated, while *atpa* was elevated, and the *kcnj12* potassium channel gene were down-regulated, the latter strongly (~6 fold).

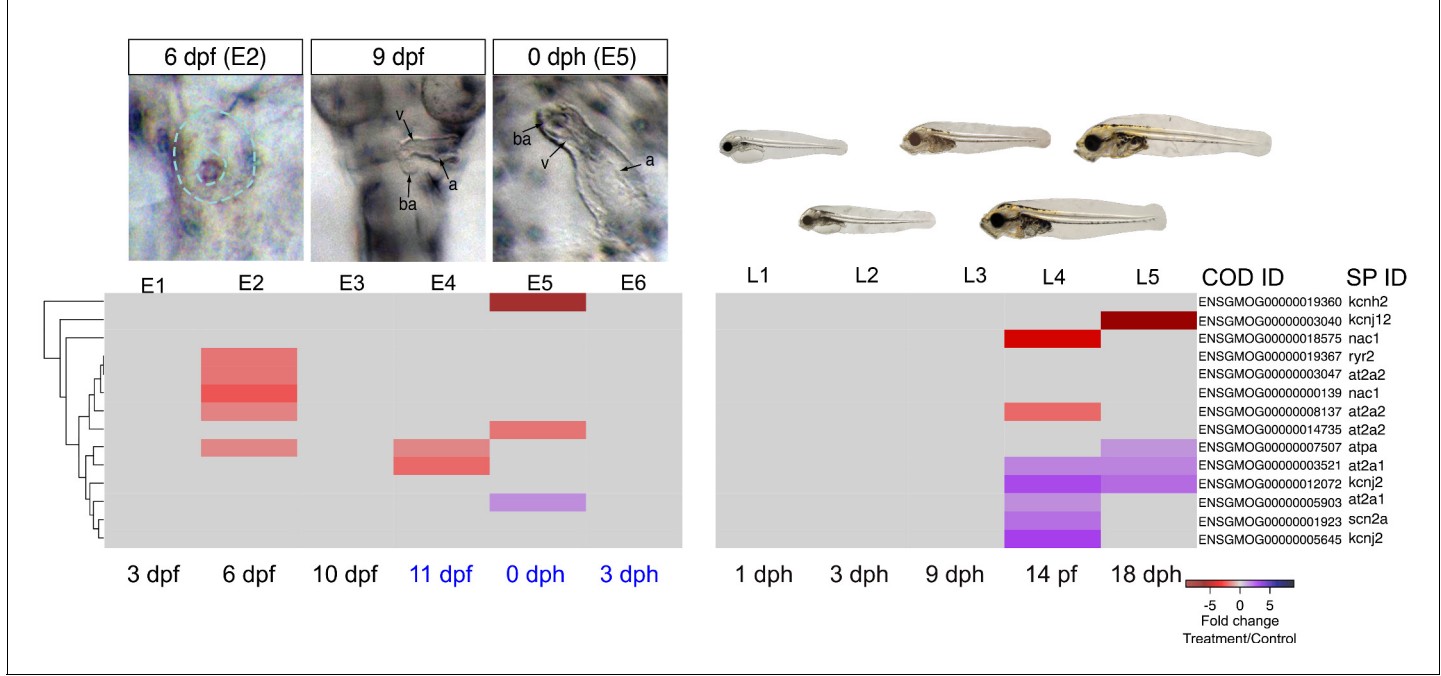

**Figure 6.** DEGs involved in E–C coupling. Embryonic developmental samples (E1–6) were collected during (black lettering) and after (blue lettering) crude oil exposure. Oil exposure was continuous across the larval sampling points (L1–5).

## Genes associated with craniofacial abnormalities

Craniofacial structures that shape the head include cartilage derived from neural crest cells and muscles that develop from paraxial mesoderm. Neural crest cells migrate from the anterior neural tube to form the pharyngeal arches with both dorsal (upper jaw) and ventral (lower jaw) patterning. They then differentiate into chondrocytes (*Knight and Schilling, 2006*; *Simões-Costa and Bronner, 2015*) and grow by processes such as convergence-extension (*Shwartz et al., 2012*; *Kamel et al., 2013*). Concurrently, mesodermal cells differentiate into patterned muscle in appropriate association with partner cartilage. Several lines of evidence suggest multidirectional signaling between all associated tissues, including endoderm (i.e. pharyngeal pouches), mesoderm, and overlying ectoderm (*Minoux and Rijli, 2010*; *Medeiros and Crump, 2012*; *Kamel et al., 2013*; *Kong et al., 2014*). Studies on zebrafish craniofacial mutants have primarily focused on the neural crest cell lineage, with less attention to muscle development or interactions between developing muscle and cartilage (*Lin et al., 2013*). Defects in oil-exposed haddock were marked by a dose-dependent reduction in more anterior cartilages (*Figure 1E–G*). This affected the basicranium most severely, with progressive loss of more posterior arch derivatives. Where present, craniofacial cartilage appeared small and distorted, without transformation to dorsal or ventral fates. This morphometry superficially aligns to several zebrafish mutants affecting either neural crest cell (*Kimmel et al., 2001*; *Nissen et al., 2006*; *Lu and Carson, 2009*; *Kamel et al., 2013*) or muscle development (*Hinits et al., 2011*; *Shwartz et al., 2012*). We therefore interpreted developmental changes in haddock gene expression in the context of these well-characterised zebrafish mutants.

The expression patterns of 12 genes with known roles in neural crest cell-dependent craniofacial development were significantly altered in the highest exposure regime (*Figure 6*). Read counts for these genes were all relatively low, consistent with highly restricted tissue-specific expression patterns (Supporting dataset 1, *Sørhus et al., 2017*). At 6 dpf, prior to visible craniofacial malformation, we observed lower expression levels of *foxi1* (pharyngeal pouches) *wnt9b* (ectoderm), *fgfr2* (chondrocytes), and *fgfr3* (chondrocytes) compared to control (*Figure 7A*). The downregulation of *wnt9b* and *fgfr2* persisted to 10 dpf, together with a downregulation of *tgfb3* and two *sox9b* paralogues, the latter also expressed in neural crest cell-derived chondrocytes. Conversely, *edn1*, *dlx3b*, and *dlx5a* were upregulated at 10 dpf. In zebrafish embryos the two *dlx* genes are normally expressed in

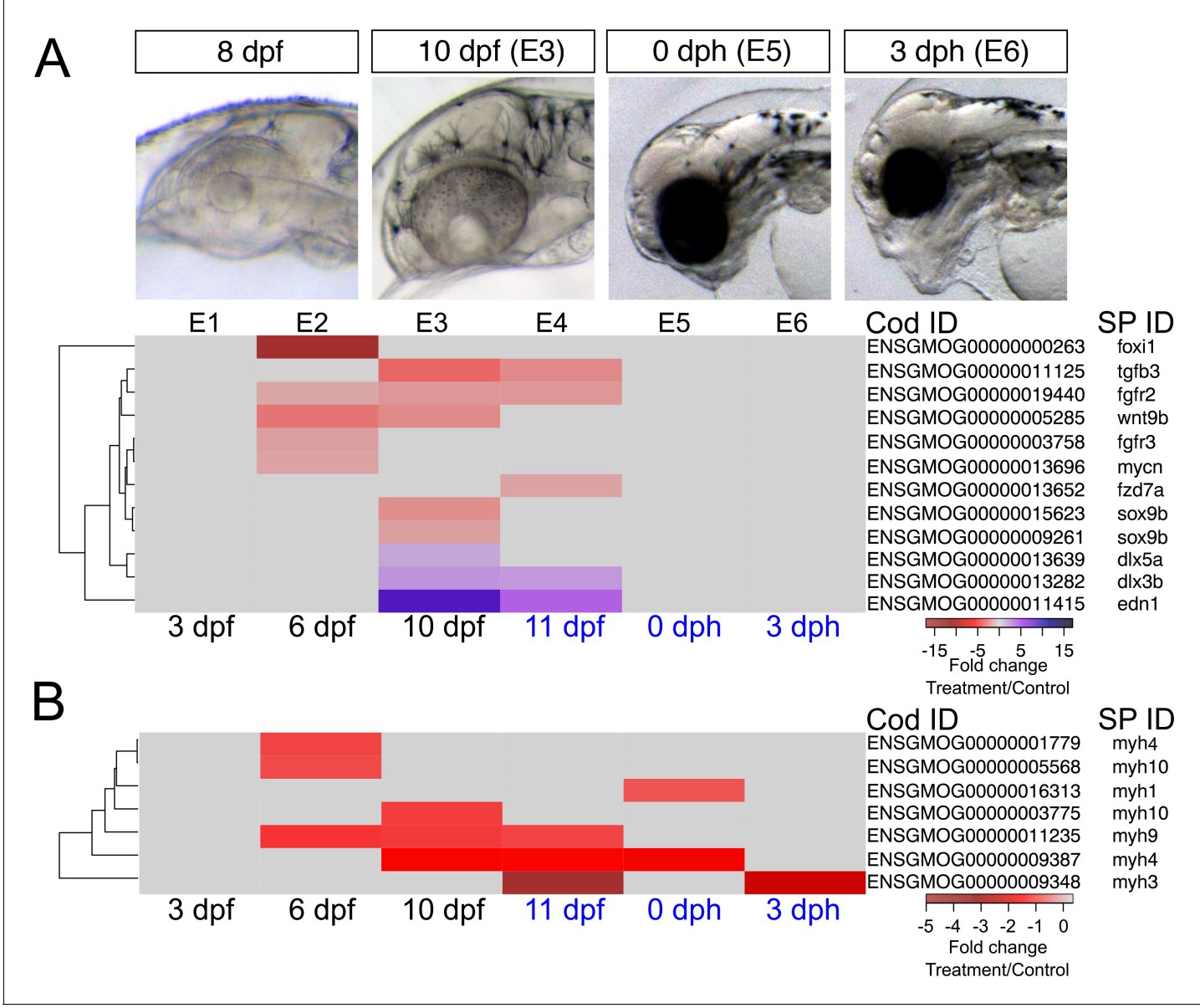

**Figure 7.** DEGs involved in craniofacial development. (a) Regulation of genes involved in craniofacial development during and after embryonic exposure. (b) Regulation of myosin heavy chain genes. Purple: increased expression, red: decreased expression in exposed group.

endoderm and arch neural crest cell-derived mesenchyme (*Talbot et al., 2010*). At 11 dpf, expression levels were down for *fgfr2, fzd7a* (a receptor for Wnt9b; chondrocytes), and *tgfb3* and up for *edn1* and *dlx3b*. None of these genes were differentially expressed after hatching (*Figure 7A*, *Supplementary file 1I*).

Genes controlling craniofacial muscle patterning are poorly characterised; however, muscle determination factors (e.g., *myod, myf6* [*Li et al., 2014*]) appeared unaffected in oil-exposed haddock. Nevertheless, expression levels were significantly reduced for genes involved in the terminal differentiation of skeletal muscle cells, including several myosin heavy chain genes (*myh*) (*Figure 7B*, *Supplementary file 1J*). These included *myh4, myh9,* and *myh10* paralogues at 6 dpf, *myh9* and another paralogue of *myh4 and myh10* at 10 dpf, and *myh3, myh4* and *myh9* at 11 dpf. Only *myh3* remained significantly downregulated after hatching (3 dph) relative to controls. Notably, expression of *myh1*, encoding the major fast myosin heavy chain gene expressed in the body musculature

(*Thisse et al., 2001*), was largely unaffected except for a small reduction at 11 dpf. Other myosin genes specific to muscle groups in the head and trunk (*Peng et al., 2002*; *Elworthy et al., 2008*), on the other hand, showed more complex differential expression patterns.

## Genes associated with ion and water regulatory imbalance

Fluid accumulation in the form of edema is a hallmark indication of crude oil toxicity in fish embryos. Although patterns of edema formation vary across freshwater and marine fish species (*Incardona and Scholz, 2016*), it nearly always involves anatomical compartments adjacent to the heart and the yolk sac. However, marine embryos are hyposmolar to surrounding seawater, and they should therefore lose water along a diffusion gradient if osmoregulation is disrupted as a consequence of heart and circulatory failure. In fish embryos and yolk sac larvae, osmoregulation is controlled by MRCs in the epidermis and yolk sac membrane that actively secrete NaCl (specifically Cl⁻) to maintain an appropriate water and ion balance (*Hiroi et al., 2005*). Genetic and pharmacologic studies have shown that circulation is required to maintain embryonic MRC cell function. For example, total body osmolality increased in seawater-adapted tilapia embryos with a ~50% reduction in total cardiac output (*Miyanishi et al., 2013*). Therefore, edema formation in marine species with oil-induced circulatory defects is not a consequence of water moving into the embryo (as it is for freshwater species) but rather water moving along an internal osmotic gradient from peripheral tissues to the vicinity of the heart and yolk sac. Accordingly, dorsal anterior finfold defects in edematous embryos and larvae (*Figure 1H,I*) represent a visible phenotypic anchor for ionoregulatory disruption, a third distinct oil-induced adverse outcome pathway.

Our analysis focused on key ionoregulatory proteins in MRCs and their associated genes, including Na⁺/K⁺ ATPase subunits (*at1* genes, e.g. *at1a1-a4*, *at1b1-b4*), a urea transporter (*ut1*), a Na⁺/K⁺/2Cl⁻ co-transporter (*s12a2*), the sodium-hydrogen exchanger Nhe3 (*slc9a3*), and a chloride channel, the latter an ortholog to the human cystic fibrosis transmembrane conductance regulator (*cftr*) (*Hirose et al., 2003*). The disruption of MRC function in oil-exposed haddock embryos corresponded to significantly lower levels of *at1a1-3, at1b2-3, ut1, s12a2, sl9a3,* and *cftr* (*Figure 8*, *Supplementary file 1K*). This downregulation primarily spanned a developmental window between 6 dpf and hatching (*Figure 8*, *Supplementary file 1K*). We also found significant transcriptional modifications of genes encoding other pumps, channels, and transporters specific to the nervous system and other tissues, including the aquaporins (*aqp* genes) that rapidly transport water across cell membranes (*Supplementary file 1L*). For example, there was a pronounced decrease in the expression of the primary neuronal water channel, *aqp4*, from 6 dpf to hatching and a strong upregulation of *aqp12* at hatch (*Figure 8*). Crude oil exposures therefore appear to cause osmotic stress in the developing embryonic nervous system.

During larval exposure, edema accumulated in different compartments from embryos, and there were corresponding differences in expression of genes related to ion and water balance. At late stages of larval exposure, edema accumulated in the peritoneal cavity, and the dorsal marginal finfold appeared increased rather than decreased as in embryos. Fewer ion transport genes were affected, with increased expression observed for only *at1a3* at 14 dph, and one *at1b2* paralogue and *at1b3* at 14 and 18 dph. Similarly, aquaporin genes were affected differently. Whereas *aqp4* was unaltered in larvae, expression of *aqp7* and *aqp9* was increased while *aqp3* and *aqp12* were decreased (*Figure 8*, *Supplementary file 1K*).

## A novel adverse outcome pathway: disruption of cholesterol homeostasis

Cholesterol is an essential structural component required for maintaining both the integrity and the fluidity of all metazoan cell membranes. Cholesterol is sourced from de novo cellular synthesis and from the uptake of external lipoprotein cholesterol from the circulation (*Bjorkhem and Meaney, 2004*). During fish development, cholesterol is mobilized from the yolk and distributed to cells during embryonic and larval yolk sac stages. Later, after the yolk is absorbed and larvae begin exogenous feeding, cholesterol is transported from the intestines. Crude-oil-exposed haddock embryos and larvae with the most severe morphological abnormalities were visibly unable to effectively mobilize yolk (*Figure 1J,K*). Moreover, larvae from the highest exposure concentration had less visible

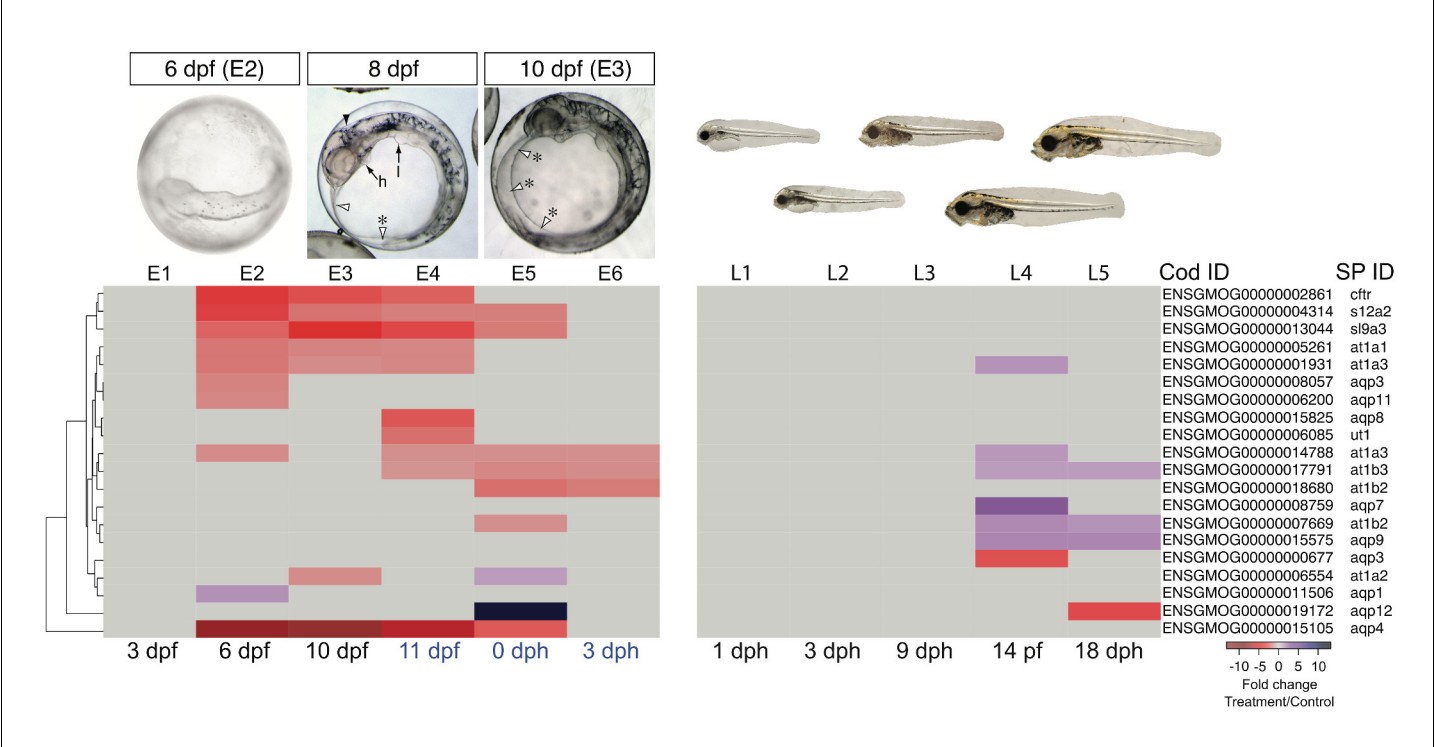

**Figure 8.** DEGs involved in osmoregulation. E1–E6: Embryonic exposure, L1–L5: Larval exposure. Black letters: during exposure, blue letters: after exposure.

food in their stomachs relative to controls. These observations together suggest that chemical components of crude oil may deprive developing tissues of externally available cholesterol.

Of the 28 genes differentially regulated at all developmental time points, 5 are involved in cholesterol synthesis and feedback control (*Table 1*). These include 3-hydroxy-3-methylglutaryl-coenzyme A reductase, an enzyme encoded by *hmdh* that plays a primary feedback regulation role in the cholesterol biosynthetic pathway (*Brown and Goldstein, 2009*). Although reduced yolk absorption was not physically measureable in exposed embryos until after 3 dph, genes controlling cholesterol synthesis were upregulated much earlier, prior to visible cardiac circulation (6 dpf/E2). We also detected complex regulation of apolipoproteins during and after exposure, with mainly down-regulation of *apob* paralogs before first heartbeat (6 dpf/E2) and up-regulation of *apoa4*, *apod apoeb* and *apoc2* after initiation of cardiac circulation (Supporting dataset 1, *Sørhus et al., 2017*). Scavenger receptor class B 1(encoded by *scarb1*), a transcytotic receptor for cholesterol-containing high-density lipoprotein (*Acton et al., 1996*), was also down-regulated in the exposed groups in haddock at 6 dpf (Supporting dataset 1, *Sørhus et al., 2017*). Pathway analysis was also consistent with a significant effect on cholesterol homeostasis (*Figure 3—figure supplement 1*).

In the larval exposure, we also detected increased expression of *hdmh*, *erg1*, *cp51a*, and *npc2* (encoding the proteins squalene epoxidase, and cytochrome P450 51A, Niemann-Pick disease, type C2, respectively) at the latest stages examined (14 and 18 dph). Conversely, pathways involved in digestion – that is, pancreatic secretion, protein digestion, and protein absorption – were suppressed. This includes the downregulation of genes encoding digestive enzymes such as trypsin and chymotrypsin (*Figure 4—figure supplement 1*). The stomachs of oil-exposed larvae at the final time point were relatively empty, and the associated loss of food-derived nutrients likely triggered the observed increase in endogenous cellular cholesterol synthesis.

## Unaltered gene expression in relation to visibly normal organs: lateral line and liver

Whereas abnormal phenotypic traits corresponded to differential gene expression, genes associated with normal traits were unchanged. For example, the lateral line and liver appeared normal in the most severely affected embryos (*Figure 1—figure supplement 1*). Consistent with this, markers for the lateral line (protein atonal homolog 1, *atoh1*) (*Cai and Groves, 2015*), liver growth (hepatocyte growth factor, *met)* and differentiation (genes encoding wnt2 and 2b protein (*wnt2, wnt2b*) (*Wilkins and Pack, 2013*), hematopoietically-expressed homeobox protein (*hhex*), and protein heg (*heg*)) (*Supplementary file 1M*) were not significantly modified. While some markers for liver differentiation, including genes encoding transferrin (*tfr1*) and fatty acid binding protein (*fa10a*) were differentially expressed, the changes were subtle and not consistent throughout development. Similarly, the related KEGG pathways that are inclusive of these genes were relatively unaffected by oil exposure at all time points. As noted above, IPA failed to identify significant enrichment of pathways related to phenotypically normal structures.

## Discussion

Overall, we observed tight anchoring of temporal gene expression patterns to measurable phenotypes in crude oil-exposed haddock. First, the global changes in gene expression observed in the embryonic and larval exposures matched the general nature and severity of phenotypes. Embryonic exposure to crude oil or component cardiotoxic PAHs produces a coarse chemical phenocopy of the loss-of-function zebrafish mutants affecting heart function or development (*Incardona et al., 2004*). Many aspects of the oil toxicity phenotype are secondary to a loss of circulation—that is, defects in non-cardiac tissues, such as the eye, that require circulation for normal organogenesis (*Incardona et al., 2004*). In contrast, larval stages are primarily a period of rapid growth after major organogenesis is complete, and the injury phenotype in larvae is less severe. Consistent with this, embryos showed a larger number of DEGs than larvae, with a preponderance of down-regulation. Second, we identified specific changes in the expression of key genes involved in the function or morphogenesis of individual tissues and organs with visible abnormalities. Given unaltered gene expression and lack of statistically enriched pathways associated with apparently unaffected structures such as the liver, the DEGs in oil-exposed haddock indicate a disruption of specific developmental processes, as opposed to non-specific effects (e.g. general developmental delay).

This study demonstrates the ability to resolve changes in tissue-specific genes in a pool of total RNA from embryos and larvae, even for organs such as the heart that contribute a very small fraction to total tissue mass. A key finding is the general correlation of read count data with tissue specific patterns previously characterized in model species. Our findings have important implications for the utility of RNA-Seq and other quantitative measures of mRNA abundance in whole embryo or larval samples. For example, this demonstrates the feasibility of developing real-time monitoring tools based on quantification of gene expression in environmental samples collected following an oil spill. In addition, our extensive manual curation of the transcriptome groundtruths the utility of applications like IPA for use with non-model, non-mammalian organisms. Moreover, the changes in gene expression identified here represent significant information to be added to existing cardiotoxicity AOPs and novel AOPs associated with disruption of osmoregulation and lipid metabolism.

Two major initiating events for crude-oil-associated cardiac defects during fish development are chemical blockade of $IK_r$ repolarizing potassium currents, (encoded by *kcnh2*) and disruption of intracellular calcium handling, the latter culminating in sarcoplasmic reticulum (SR) calcium depletion through effects on either RyR or SERCA2 (encoded by *ryr2* and *at2a2*, respectively) (*Brette et al., 2014*). In the fully formed heart, these pharmacologic effects impair cardiac function by inducing arrhythmia and reducing contractility (*Incardona et al., 2009*, *2014*). However, rhythm and contractility defects during heart development lead to morphological defects (*Andrés-Delgado and Mercader, 2016*). In haddock embryos, these include poor chamber looping and outgrowth of the ventricle (*Sørhus et al., 2016b*). Our data demonstrate a transcriptional cascade that is tightly linked to these defects in cardiac function (cardiomyocyte intracellular calcium cycling) and form (heart chamber growth) through *bmp10*.

While chemical blockade of calcium cycling alone would be sufficient to induce the ventricular arrhythmias observed in oil-exposed embryolarval haddock, other elements of the E-C coupling

physiological cascade were also selectively modified at the mRNA level. As shown previously using qPCR (*Sørhus et al., 2016b*), RNA-seq revealed a downregulation of genes encoding the Na/Ca exchanger (*nac1*) and IK$_r$ (*kcnh2*) in haddock embryos. Notably however, *kcnh2* downregulation was only observed at later time points, in response to the highest oil exposures that caused the most severe phenotypes. There was no consistent decrease in the mRNA for a larger suite of proteins involved in cardiac E-C coupling. Assuming a broader transcriptional response in the heart was not masked by more abundant, normal expression of these genes in larger non-cardiac tissues, other non-specific mechanisms were unlikely to contribute to the formation of misshapen hearts. Moreover, the changing expression of key E-C coupling genes is a close match to the cardiac arrhythmia phenotype in both embryos and larvae. This includes a marked downregulation (>5 fold) of *kcnj12*, which encodes a subunit of the repolarizing IK$_1$ current and causes the same types of ventricular arrhythmias as a reduction of *kcnh2* (*Domenighetti et al., 2007*). At the same time, the up-regulation of E-C coupling genes following the chemical induction of arrhythmia in the larval exposures suggest that the more mature larval heart mounts a compensatory response.

Intracellular calcium has multiple direct roles in regulating gene expression, including the process of excitation-transcription (E-T) coupling (*Wamhoff et al., 2006*). Our findings suggest that E-T coupling may link reduced cardiomyocyte calcium cycling to structural defects in the haddock heart. Among vertebrates, *bmp10* is expressed exclusively in the early tubular hearts of zebrafish, mouse, and chick embryos. The normal function of Bmp10 in the developing heart is primarily to drive ventricular cardiomyocyte proliferation during trabeculation (*Grego-Bessa et al., 2007*), a relatively late process during embryogenesis (around hatching stages in fish). Both loss of and excess Bmp10 leads to severe abnormalities in ventricular development in mouse (*Chen et al., 2004*). In mice lacking the RyR-associated Fkbp12 protein, disruption of SR calcium handling leads to ventricular defects through elevated *bmp10* transcription (*Shou et al., 1998*; *Chen et al., 2004*), probably through calcium-dependent activation of myocardin (*Wamhoff et al., 2004*, *2006*), the transcriptional activator of *bmp10* (*Huang et al., 2012*). Moreover, Bmp10 is the most potent Bmp family member, showing greater resistance to Bmp antagonists (e.g. Noggin) than Bmp4 (*Lichtner et al., 2013*), the primary cardiac Bmp family member at early stages. While *bmp10* normally functions at late stages of cardiogenesis, *bmp4* is normally expressed at the cardiac cone stage in zebrafish. At this stage, *bmp4* levels shift from radially symmetric to elevated on the left side of the cone, to drive proper looping (*Chen et al., 2004*). Loss of this asymmetry with ectopic *bmp4* leads to un-looped hearts. Therefore, the premature up-regulation of a more potent family member, *bmp10*, at the cone stage is very likely to underlie the looping defects observed in oil-exposed embryos. Further evidence for *bmp10* overexpression initiating abnormal cardiac morphogenesis is the secondary up-regulation of the Bmp10 target gene *nkx25* (*Chen et al., 2004*). In zebrafish, *nkx25* overexpression or loss of function (*Tu et al., 2009*) yields a reduced ventricle, and *nkx25* must be down-regulated or antagonised in specific regions of the ventricle in order to form specialised conduction cells through the repressor action of Tbx3 (*Hoogaars et al., 2004*). The higher levels of *tbx3* that follow upregulation of *nkx25* and subsequent downregulation of *anf* thus likely reflect an imbalance between myocardial and non-myocardial cell fates within the ventricle. Thus, normal heart development in zebrafish requires tight control over *bmp10*, *nkx25*, and *tbx3* expression, and all three genes were dysregulated in oil-exposed haddock. The observed ventricular and looping defects may represent chemical phenocopies of the *fkbp12* mutant, wherein reduced intracellular calcium transients are linked to altered *bmp10* expression by E-T coupling, thereby changing cell fate and chamber formation in the developing heart. Calcium-mediated E-T coupling may also be a feedback mechanism for altering the expression of genes that encode repolarizing potassium channels.

Although the haddock with craniofacial deformities superficially resemble certain zebrafish mutants, associated changes in gene expression suggest a more complex developmental perturbation than previously described. As is the case with Bmp10, Edn1 is a strong morphogen that must be tightly regulated, as both too much and too little lead to craniofacial defects (*Sato et al., 2008*; *Clouthier et al., 2010*). Higher levels of *edn1* observed here are thus highly likely to be related to the craniofacial defects, which is supported by the subsequent up-regulation of its target *dlx* genes. However, the phenotype does not appear to reflect changes in dorsal-ventral patterning, as expected for perturbation of *edn1*-dependent NCC identity. Most studies of craniofacial development in zebrafish and other vertebrates have focused on NCC-derived cartilaginous precursors. However, craniofacial skeletal elements develop in synchrony with their associated muscles

(*Schilling and Kimmel, 1997*), and defects in muscle formation or function produce phenotypes that are in many ways indistinguishable from primary cartilage defects and appear more similar to the phenotypes observed here (*Hinits et al., 2011*; *Shwartz et al., 2012*). Mesodermal precursors of craniofacial muscle cells express *edn1*, the downregulation of which is associated with terminal muscle differentiation (*Choudhry et al., 2011*). A failure to downregulate *edn1* is consistent with failure to up-regulate or maintain *myh* gene expression. It is unknown whether a failure of skeletal muscle to terminally differentiate would lead to reduced expression of genes associated with NCC development and cartilage growth and survival—that is *foxi1*, Wnt pathway genes (*wnt9b*, *fzd7a*, *mycn*), Fgf receptors and *tgfb3*. On the other hand, the early expression of *foxi1* has no clear linkage to craniofacial muscle development in the literature, but it is required for NCC survival indirectly by driving Fgf signaling (*Edlund et al., 2014*). Sorting out the precise pathways of craniofacial malformation will thus require a concerted spatial localization of these DEGs.

The regulation of genes controlling ion and water balance, combined with the collapse of the dorsal marginal finfold, provides a novel insight into the origins of edema in marine fish. Conversely, the genetic elements of this phenotype provide a starting point to study the molecular basis of buoyancy control in pelagic fish larvae. Our findings suggest that crude oil may disrupt MRC function, leading to decreased expression of MRC channel and transporter genes. These effects could be direct, indirect, or both. First, ion and water balance in embryos require cardiac circulation (*Miyanishi et al., 2013*). Flow is essential for osmoregulatory counter-current exchange in the gills and kidney (*Somero, 1998*; *Grosell, 2011*), and similar principals should apply to the yolk sac epidermis. Second, epidermal MRCs are likely directly exposed to the highest PAH concentrations, because the highest Cyp1a upregulation in response to oil occurs in the skin of embryos and yolk sac larvae (*Sørhus et al., 2016b*). PAHs or their metabolites could block solute carriers in a similar manner as shown for cardiac calcium and potassium channels (*Brette et al., 2014*; *Incardona et al., 2014*). Lastly, MRC function could be impaired if the metabolic cost of PAH degradation competes with ion transport. Although the effects of oil exposure on salt and water balance in fish embryos have not been examined previously, exposing water-soluble oil fraction to adult Pacific herring (*Clupea pallasi*) showed an effect on ion homeostasis (*Kennedy and Farrell, 2005*).

Genes involved in ion and water balance were differentially regulated in embryos and larvae, in close correspondence to the loss and expansion of the dorsal subdermal space, respectively. Edema flows along the path of least resistance and accumulates in expandable spaces. In haddock embryos, fluid moves from the dorsal subdermal space to the yolk sac. At the larval stage, the presence of fluid in the dorsal finfold suggests that a developed peritoneal cavity and body wall provide greater resistance than the thin, permeable yolk sac membrane. As a consequence, the central nervous system is likely deprived of water during embryonic development and turgidly stressed at the larval stages. Changes in cell volume can modify intra- and extracellular ionic concentrations, and thus the electrophysiological properties of neurons (*Pasantes-Morales et al., 2000*). This may underlie the observed down-regulation of the main water channel in the brain, aquaporin 4 (*aqp4*), during and after embryonic exposure. Although not a focus of the current study, there are likely important mechanistic connections between a loss of embryonic MRC function, disrupted osmoregulation in the brain, and pathophysiological impacts on neuronal development.

Our findings also provide new insights into fundamental transcriptional mechanisms of lipid mobilization from yolk. The delivery of yolk-derived cholesterol not been widely studied in early fish embryos, which are distinct from other vertebrates in that the yolk mass is contained separately from the vasculature by the yolk syncytial layer (YSL, or periblast in older literature). The YSL is a multicellular membrane that forms during gastrulation and has been studied mostly for its role in early pattern formation (*Carvalho and Heisenberg, 2010*). At later embryonic and larval stages, the YSL transports yolk-derived nutrients into the circulation (e.g, [*Poupard et al., 2000*]). Although cholesterol is exported from the yolk prior to established circulation (*Fraher et al., 2016*), the cellular basis for this is not known—for example, by trancytosis or direct membrane transport. Cellular sterol levels are tightly controlled by membrane-bound transcription factors, which are cleaved and activated when membrane cholesterol levels fall, leading to transcription of *hmdh*, the rate-limiting enzyme for cholesterol synthesis (*Brown and Goldstein, 2009*). Therefore, the brisk up-regulation of intrinsic cholesterol biosynthetic genes, especially *hmdh*, prior to a visible mobilization of yolk, indicates the importance of yolk-derived cholesterol for embryonic tissues. In zebrafish and other fish, apolipoprotein genes (e.g. ApoB and ApoE) as well as Scarb1 are first expressed in YSL (*Poupard et al., 2000*;

*Thisse et al., 2001*; *Otis et al., 2015*). The downregulation of multiple *apob* paralogs and *scarb1* during early development suggest a specific defect in packaging and transporting of lipoprotein-cholesterol in the YSL, possibly involving Scarb1-dependent transcytosis. At stages subsequent to heartbeat initiation, the upregulation of intrinsic cholesterol biosynthetic genes likely reflects cholesterol deprivation as a consequence of the heart failing to deliver lipoproteins from the yolk (embryos) and intestine (larvae). The identification of broader lipid metabolism pathways by IPA also suggests that oil exposure leads to more global derangements relating to either poor yolk absorption or dysfunction of the YSL. It is well known both that embryonic oil exposure leads to later growth impairment (e.g. [*Heintz, 2000*; *Incardona and Scholz, 2016*]) and that lipids provide critical fuel for marine fish larvae, particularly at the first-feeding stage (*Tocher et al., 2003*). The consequences of disordered lipid metabolism for larval physiology and survival should be a focus for future studies. Importantly, the induction of cholesterol synthetic genes is a promising and novel indicator of crude oil toxicity to fish embryos.

Finally, we consider the four individual genes that were consistently upregulated across all developmental stages: *ccbe1*, *rhag*, *foxq1*, and *fgf7*. Elevated tissue pressure is a signal for lymphangiogenesis (*Schulte-Merker et al., 2011*) and the marked increase in *ccbe1* expression is consistent with a compensatory formation of lymphatics to alleviate the physical effects of edema (*Planas-Paz et al., 2012*). In zebrafish, the secreted Ccbe1 protein appears to function exclusively in lymphangiogenesis (*Hogan et al., 2009*; *Le Guen et al., 2014*) by enhancing the activation of VEGF-C (*Le Guen et al., 2014*). For future spills, quantitative measures of *ccbe1* upregulation should serve as very sensitive indicators of edema formation in crude oil-exposed fish embryos.

The Rh proteins are primarily structural components of erythrocyte membranes but were also recently identified as ammonia transporters (*Weiner and Verlander, 2014*). The *rhag*, *rhbg*, and *rhcg* genes are important for excretion of nitrogenous waste in fish (*Braun et al., 2009*). The strong upregulation of *rhag* observed in haddock embryos and larvae might be a consequence of MRC dysfunction. The increase in *rhag* expression corresponded to a downregulation of urea transporters *ut1* and *ut2* in embryos but not larvae, suggesting metabolic defects relating to amino acid or protein catabolism and an increased demand for nitrogen excretion. Alternatively, *rhag* may play a novel role in embryolarval physiology.

The last two highly responsive genes, *foxq1* and *fgf7,* may be linked to the craniofacial injury phenotype based on prior work in other species. In zebrafish and frog embryos, *foxq1* is expressed in the pharyngeal region (*Choi et al., 2006*; *Planchart and Mattingly, 2010*). In chick embryos, *fgf7* is first expressed in the pharyngeal endoderm and head mesoderm (*Kumar and Chapman, 2012*). However, *foxq1* has been shown to be a downstream target of AhR in other tissues (*Faust et al., 2013*), and crude oil exposures strongly induced *fgf7* expression in the livers of juvenile polar cod (*Andersen et al., 2015*). Both genes are promising new biomarkers for future studies, but tissue localization during craniofacial development in embryonic fish is needed to confirm a role in this adverse outcome pathway. In zebrafish exposed to the dioxin TCDD, *foxq1* was up-regulated in the branchial arches (*Planchart and Mattingly, 2010*), but TCDD-induced craniofacial defects have been shown to be entirely secondary to cardiotoxicity (*Lanham et al., 2014*).

Lastly, our findings show how transcriptomics can inform chemical genetics and environmental forensics in non-model organisms. First, we used known spatial mRNA distributions in model species (primarily zebrafish) to more accurately phenotypically anchor the transcriptome data for crude oil-exposed haddock. This accelerates the pace of discovery, particularly given difficulties in applying zebrafish methods for *in situ* hybridization to wild species. Second, our results reveal transcriptional aspects of chemical injury in fish, in response to a global pollution threat. Although endocrine disrupting compounds act on steroid hormone receptors or epigenetically modify DNA or histones (*Walker and Gore, 2011*), it has been much less clear whether crude oil, which acts on protein targets (e.g. [*Brette et al., 2014*]), also influences mRNA levels as part of a widely studied developmental injury phenotype.

In conclusion, our findings greatly expand our understanding of crude oil impacts to fish early life stages at a molecular level. The scientific focus on highly vulnerable fish embryos and larvae began with the 1989 *Exxon Valdez* disaster in Prince William Sound, Alaska. Major crude oil spills have continued worldwide—for example, the 2002 *Prestige* spill in Spain, the 2007 *Hebei Spirit* spill in Korea, and the 2010 *Deepwater Horizon* blowout in the northern Gulf of Mexico. An enduring challenge spans these and other spills: namely, the accurate determination of fisheries losses as a consequence

of oiled spawning habitats. Our results more clearly define the different categories of developmental injury. We have also identified responsive genes that hold promise as sensitive molecular indicators of physiological stress. These can be developed into novel assessment tools for diagnostic use in future spill zones. Lastly, differential patterns of gene expression in oil-exposed haddock provide insight into fundamental but still poorly understood developmental processes in marine fish, including calcium-mediated excitation-transcription coupling, fluid balance, lipid mobilization, and buoyancy.

## Materials and methods

For detailed procedures on animal collection, maintenance, exposure regime and analytical chemistry see (*Sørhus et al., 2015*, *2016b*). All animal experiments within the study were approved by NARA, the governmental Norwegian Animal Research Authority (http://www.fdu.no/fdu/, reference number 2012/275334-2). All methods were performed in accordance with approved guidelines.

### Animal collection, maintenance, and exposure set-up and regime

The samples transcriptome profiled here are the exact same samples described in *Sørhus et al. (2016b)*, with the methodology for animal collection, maintenance, and crude oil exposure provided therein. Briefly, fertilized eggs were collected from tanks with wild captured Atlantic haddock (collected spring 2013) and transferred to indoor egg incubators (7°C). At experiment start, eggs from the egg incubators were transferred to 50 L exposure tanks (7.8°C). Embryonic and larval stages of Atlantic haddock were exposed to two doses, low and high dose, in addition to a pulsed high dose (see *Figures 3* and *4* for details). Heidrun oil blend was artificially weathered by distillation and then pumped into the dispersion system using a HPLC pump (Shimadzu, LC-20AD Liquid Chromatograph Pump) with a flow of 5 µL/min together with a flow of seawater of 180 mL/min. The system described in Nordtug et. al 2011 (*Nordtug et al., 2011*) generates an oil dispersion with oil droplets in the low µm range with a nominal oil load of 26 mg/L (stock solution). Water samples were collected from each exposure tank at the beginning of each experiment and subjected for detailed PAH analysis. At end of exposure, pooled samples of eggs and larvae were extracted by solid liquid extraction and purified by solid phase extraction prior to analysis by GC-MS/MS (*Sørensen et al., 2015*) to reveal the PAH content in animals. After end of exposure, the animals were transferred to new tanks with clean water. Images of the animals were collected from 12 and 8 stages during and after embryonic and larval exposure, respectively. Videos from 60 (embryonic) and 48 (larval) individual embryos/larvae per treatment per stage were collected from 9 dpf, 0 dph, and 3 dph (embryonic exposure) and 2 dph and 9 dph (larval exposure). The experiments included four replicate tanks for each dose, and pooled samples from three replicate tanks for each dose were subjected for sequencing (see details below).

### Total RNA and cDNA preparation

Total RNA was isolated from frozen pools of embryos and larvae using Trizol reagent (Invitrogen) per manufacturer instructions. This included a DNase treatment step using a TURBO DNA-*free* kit (Life Technologies Corporation). RNA was quantified using a Nanodrop spectrophotometer (Nano-Drop Technologies), and confirmed using a 2100 Bioanalyzer (Agilent Technologies). cDNA was subsequently generated using SuperScript VILO cDNA Synthesis Kit (Life Technologies Corporation), according to the manufacturer's instructions. The cDNA was normalized to obtain a concentration of 50 ng/µL.

### Real-time qPCR

Six responsive genes from the transcriptome were validated by real-time quantitative PCR (qPCR) (*Figure 3—figure supplement 2* [Embryonic exposure], *Figure 4—figure supplement 2* [Larval exposure]). Specific primers and probes (*Supplementary file 1N*) for a reference gene (*ef1α*) and the six DEGs (*cp1a*, *wnt11*, *kcnh2*, *nac1*, *cac1c,* and *at2a2*) were designed with either Primer Express Software (Applied Biosystems) or Eurofins qPCR probe and primer design software (Eurofins Scientific), according to the manufacturer's guidelines. The two methods generally yielded the same quantitative trends. Primer and probe sequences are shown in *Supplementary file 1J*. TaqMan PCR assays were performed in duplicate, using 384-well optical plates on an ABI Prism Fast 7900HT

Sequence Detection System (Applied Biosystems) with settings as follows: 50°C for 2 min, 95°C for 20 s, followed by a 40 cycles of 95°C for 1 s and 60°C for 20 s. For each 10 µL PCR reaction, a 2 µL cDNA 1:40 dilution was mixed with 200 nM fluorogenic probe, 900 nM sense primer, and 900 nM antisense primer in 1xTaqMan Fast Advanced Master Mix (Applied Biosystems). Gene expression was calculated relative to the exposure time zero sample (2 dpf and 0 dph in embryonic and larval exposures, respectively) using the $\Delta\Delta\Delta$Ct method, generating reference residuals (*Edmunds et al., 2014*) from *ef1a* and *at2a2*.

## Extraction of mRNA, RNA sequencing, and bioinformatics

cDNA library preparation and sequencing was performed by the Norwegian Sequencing Centre (NSC, Oslo, Norway) using the Illumina TruSeq RNA Sample Preparation Kit. A total of 132 samples were sequenced and 126 were subjected for analysis: Three (control, low and high dose) or two (pulse) biological replicates for each dose from six stages during and after embryonic exposure and three biological replicates for each dose from five stages during larval exposure (see *Figures 3* and *4*). Paired-end libraries were sequenced on the Illumina HiSeq2500 platform. The raw data are available from the Sequence Read Archive (SRA) at NCBI (Accession ID: PRJNA328092).

The high sequence similarity between the two species justified use of the cod template, and we chose a verified model over a reference-free de novo transcriptome approach to avoid fragmentation noise and false positives from un-collapsed genes. The average sequence similarity between mapped haddock reads and the previously verified cod genome was 98.4%. Moreover, of the 20,954 annotated cod genes, there were 18,990 (90.6%) corresponding haddock genes with at least 10 reads in one sample. Thus, the RNA sequencing data were mapped to the coding sequences of the cod genes (*Star et al., 2011*) using the Bowtie aligner (RRID:SCR_005476) (*Langmead et al., 2009*) and annotated as described in *Sørhus et al. (2016a)*. Samtools idxstat (RRID:SCR_002105) (*Li et al., 2009*) was used to extract the number of mapped reads which were then normalised to the total number of mapped sequences. NOISeqBIO (RRID:SCR_003002) (*Tarazona et al., 2011, 2015*) was used to identify differentially expressed genes (DEGs, threshold of 0.95). The total number of reads averaged 41.39 million per sample and the mapping efficiency averaged 32.69%, giving an overall average of 13.51 million mapped paired end reads (125 bp) for each sample. Given the absence of untranslated regions (UTRs) and mitochondrial genes from the reference cod genome, reads with a UTR sequence and all reads for mitochondrial genes were excluded from the haddock analyses.

Only genes with 10 reads or more in at least one of the samples were included for further analysis. Kyoto Encyclopedia of Genes and Genomes (KEGG) pathways analysis (RRID:SCR_012773) (*Kanehisa et al., 2012*) was performed by mapping the KEGG annotated DEGs from NOISeqBIO to KEGG pathways as described in the KEGG Mapper tool. Heat maps were generated from fold change data in R (*R Core Team , 2013*) and Venn diagrams were created using the web-tool, Venn (http://bioinformatics.psb.ugent.be/webtools/Venn). Individual genes involved in cardiac and craniofacial development, osmoregulation and lipid metabolism, or not directly linked to KEGG pathways were curated manually in an intensive process that took a full year. In a previous effort, we characterized the transcriptome of normally developing haddock embryos and larvae (*Sørhus et al., 2016a*). Through extensive literature searches (PubMed, Web of Science, and Google Scholar) and reading, lists of roughly 150 key genes involved in cardiac and craniofacial development and cardiac function were assembled. After obtaining the oil exposure RNA-Seq dataset, these lists were expanded by further literature searches. Genes of interest were identified as ones that showed strong dose-dependency and had potential phenotypic association based on the literature. We relied heavily on the expression database at the Zebrafish Information Network (www.zfin.org) to determine whether individual genes of interest were expressed in the relevant tissues at the appropriate developmental stage to be associated with phenotypes, helping to narrow in on key linkages. This was performed both by searching individual gene names and by generating lists of all genes expressed in a specific tissue at a time relevant to the phenotypes (e.g. all genes expressed in the ethmoid plate between segmentation and hatch). These lists were cross-referenced to the list of DEGs to identify candidates. If zfin.org lacked expression data, we searched for any papers describing tissue localization by *in situ* hybridization in other model or non-model species.

Qiagen's Ingenuity Pathway Analysis (IPA version 01–04) (RRID:SCR_008653) was used subsequent to our manual curation. The fold change and p values were extracted from the original

NOISeqBIO output. Fold change values were multiplied by −1 to flip the direction (opposite the convention used by IPA), and any fold change values between −1 and 1 was set to 1, as non-significant values and those below 10 reads were originally set to 0.5 and 0.75. The cod genes were then BLASTed (version 2.3.0+) against the ENSEMBL zebrafish (GRCz10) and human (GCRh38) databases. The top match for each gene, filtered based on e-value (cutoff $10^{-5}$), was used to build a mapping table of the genes to the IPA database. For genes that lacked mapping at that point, SwissProt information was used to manually map. The mapping information was then combined with the fold change and p-value data and uploaded to IPA. This resulted in 17608 of 20954 genes (84%) mapping to the IPA database. Data were then analyzed using a fold change cutoff of 2.0 for both up- and down-regulation, considering relationships only when confidence was equal to the experimentally observed level. For this analysis, IPA uses a built-in right-tailed Fisher Exact Test to calculate p-values for significant functions and pathways. After each time point was analyzed individually in this manner, data outputs were used in comparison analyses in which the control dataset was compared to each dose at each time point. The output of the comparison analyses included rankings of top pathways in a variety of categories and subcategories. These rankings were expanded in the IPA software interface and inspected manually to generate *Table 3* and *Supplementary file 1B* and *1C*.

**Supporting dataset 1:** Expression of all genes in control, low dose, pulse dose, high dose during and after embryonic exposure. Highlighted FC: Prob $\geq$ 0.95; Not highlighted FC: Prob 0.8–0.95; FC = 0.5: Not significant; FC = 0.75: Less than 10 reads in both treatment and control. SP, swissprot; GB, genebank; E1, 3 dpf; E2, 6 dpf; E3, 10 dpf; E4, 11 dpf; E5, 0 dph; E6, 3 dph; C, control; L, low dose; P, pulse dose; H, high dose; FC; fold change; prob; probability BioNoiseq. (*Sørhus et al., 2017*).

**Supporting dataset 2:** Expression of all genes in control, low dose, pulse dose, high dose during larval exposure. Highlighted FC: Prob $\geq$ 0.95; Not highlighted FC: Prob 0.8–0.95; FC = 0.5: Not significant; FC = 0.75: Less than 10 reads in both treatment and control. SP, swissprot; GB, genebank; L1, 1 dph; L2, 3 dph; L3, 9 dph; L4, 14 dph; L5, 18 dph; C, control; L, low dose; P, pulse dose; H, high dose; FC; fold change; prob; probability BioNoiseq. (*Sørhus et al., 2017*).

## Acknowledgements

We acknowledge Michal Rejmer, Domagoj Maksan, Ørjan Karlsen and Terje van der Meeren for providing fish husbandry. Funded by the Research Council of Norway (Project no. 234367, www.forskningsradet.no), the VISTA foundation (Project no. 6161, www.vista.no) and the Institute of Marine Research, Norway. The University of Oslo's Norwegian Sequencing Centre (NSC; http://www.sequencing.uio.no) created the library and sequenced the transcriptome. Funding organizations had no role in study design, data collection and analysis, or manuscript preparation.

## Additional information

### Funding

| Funder | Grant reference number | Author |
| --- | --- | --- |
| Norges Forskningsråd | Project no. 234367 | Elin Sørhus<br>John P Incardona<br>Tomasz Furmanek<br>Nathaniel L Scholz<br>Sonnich Meier<br>Rolf B Edvardsen<br>Sissel Jentoft |
| VISTA Foundation | Project no. 6161 | Elin Sørhus |
| Institute of Marine Research | Project no. 14236 | Elin Sørhus<br>Tomasz Furmanek<br>Sonnich Meier<br>Rolf B Edvardsen |

The funders had no role in study design, data collection and interpretation, or the decision to submit the work for publication.

## Author contributions

ES, Conceptualization, Data curation, Formal analysis, Validation, Investigation, Visualization, Methodology, Writing—original draft, Writing—review and editing; JPI, Conceptualization, Data curation, Formal analysis, Supervision, Funding acquisition, Investigation, Visualization, Methodology, Writing—original draft, Writing—review and editing; TF, Resources, Software, Formal analysis; GWG, Formal analysis, Writing—review and editing; NLS, Writing—original draft, Writing—review and editing; SM, Conceptualization, Data curation, Funding acquisition, Investigation, Project administration; RBE, Conceptualization, Software, Supervision, Funding acquisition, Validation, Investigation, Methodology; SJ, Conceptualization, Funding acquisition

## Author ORCIDs

Elin Sørhus, http://orcid.org/0000-0003-3542-4201

## Ethics

Animal experimentation: All animal experiments within the study were approved by NARA, the governmental Norwegian Animal Research Authority (http://www.fdu.no/fdu/, reference number 2012/275334-2). All methods were performed in accordance with approved guidelines.

# Additional files

### Supplementary files

• Supplementary file 1. Tables. (A) Read count data for a selection of genes expressed in distinct tissues. (B) Ten most regulated Ingenuity Pathway Analysis (Categories: Development, Lipid metabolism, Molecular transport) in high dose at stages E1-E6. #, number of molecules (C) Ingenuity Pathway Analysis. Top five pathways in low, pulse and high dose at all time points during and after embryonic exposure in the categories: Top canonical pathways, Molecular and cellular functions, Physiological system development and function and Cardiotoxicity. #, number of molecules (D) Ten most up- and down-regulated genes. Genes that are represented among the ten most at more than one stage are collapsed into one row. SP, swissprot; E1-E6, embryonic exposure; L1-L5, larval exposure; PMID, PubMed identification. (E) Regulation of differentially expressed genes involved in cardiogenesis SP, swissprot; GB, genebank; FC, fold change; E1-E6, embryonic exposure; C, control; H, high dose. (F) Manually curated list of genes involved in cardiac development and function and craniofacial development and bone and cartilage maintenance. SP, swissprot; GB, genebank; Ref, references; PMID: National Center for Biotechnology Information (NCBI) PubMed identification. (G) List of excitation contraction coupling genes examined. SP, swissprot; GB, genebank. (H) Regulation of differentially expressed genes involved in excitation contraction coupling in exposed haddock. SP, swissprot; GB, genebank; FC, fold change; E1-E6, embryonic exposure; C, control; H, high dose; L1-L5, larval exposure. (I) Regulation of differentially expressed genes involved in craniofacial development. SP, swissprot; GB, genebank; FC, fold change; E1-E6, embryonic exposure; C, control; H, high dose. (J) Regulation of differentially expressed myosin heavy chain genes. SP, swissprot; GB, genebank; FC, fold change; E1-E6, embryonic exposure C, control; H, high dose. (K) Regulation of differentially expressed key genes involved in osmoregulation. SP, swissprot; GB, genebank; FC, fold change; E1-E6, embryonic exposure; C, control; H, high dose; L1-L5, larval exposure. (L) Differentially expressed genes involved in osmoregulation. SP, swissprot; GB, genebank; IE, increased expression; DE, decreased expression. (M) Genes involved in liver and lateral line development. SP, swissprot; GB, genebank. (N) Primers and probes for real time qPCR. SP, swissprot.

### Major datasets

The following datasets were generated:

| Author(s) | Year | Dataset title | Dataset URL | Database, license, and accessibility information |
| --- | --- | --- | --- | --- |
| Sørhus E, Incardona JP, Furmanek T, Goetz GW, Scholz | 2017 | Data from: Novel adverse outcome pathways revealed by chemical genetics in a developing marine fish | http://dx.doi.org/10.5061/dryad.28d2p | Available at Dryad Digital Repository under a CC0 Public |

| | | | | Domain Dedication |
|---|---|---|---|---|
| NL, Meier S, Edvardsen RB, Jentoft S | | | | |
| Sørhus E, Incardona JP, Meier S, Scholz NL, Goetz GW, Furmanek T, Edvardsen RB, Jentoft S | 2016 | Sequence data | http://www.ncbi.nlm.nih.gov/sra/PRJNA328092 | Publicly available at the NCBI Sequence Read Archive (accession no: PRJNA328092) |

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
