## [Decision Letter]

Thank you for submitting your article "Novel adverse outcome pathways revealed by chemical genetics in a developing marine fish" for consideration by *eLife*. Your article has been reviewed by three peer reviewers, and the evaluation has been overseen by a Reviewing Editor and Marianne Bronner as the Senior Editor. The reviewers have opted to remain anonymous.

The reviewers have extensively discussed the reviews with one another and the Reviewing Editor has drafted this decision to help you prepare a revised submission.

As you will see from the individual reviews below, all of the reviewers had extensive issues with the analysis and lack of detail in the description of method. Even more importantly, however, they also felt that the there was a great deal of over-interpretation and "overselling of the work", rendering many statements unsupported by the data. We feel that this is fixable but will require considerable rewriting and re-review. If you feel you are able to revise the paper accordingly, we would be willing to consider a significantly revised version for re-review.

Reviewer #1:

In this article the authors expand on previous experiments, where developing haddock are exposed to low levels of crude oil, to test for impacts on sensitive developmental endpoints. Developmental impacts are described in detail in previous reports, and this study reports transcriptomic changes during development following three oil exposure regimes.

I have two main difficulties with this paper. First, there is insufficient description of methods for gene expression analyses. Second, there is much speculation and unsupported statements such that much of the manuscript reads as over-statement.

Gene expression analysis:

Though the experimental design includes three different exposure regimes, specific gene transcriptional responses are not correlated in any way to dosing regime (high, low, pulse). The authors ought to show gene expression dose responses, at least for the genes for which they are trying to build cause-effect relationships with adverse outcomes. More convincing arguments for cause-effect would be supported by data showing that dysregulation of the gene precedes the appearance of the phenotype, and that dysregulation is more pronounced in treatments with more pronounced phenotypic effects (e.g., higher doses). The lack of detail reported here could make a reader concerned that perhaps interpretation of gene-specific responses are not as clear-cut as represented in the text. Are Figure 4, Figure 5, Figure 6, Figure 7 showing data from any dose, or just high dose? Furthermore, there is no description of how emergence of phenotypes varied with dosing regime. Text (starting paragraph two of the Results section) that describes developmental phenotypes does not distinguish between effects that differ between doses, and all of the images from exposed animals in Figure 1 appear to be only from the high dose.

Other aspects of gene expression analysis that are problematic:

There is no description of the statistical model. What was compared to what? How many tests were there? Was there multiple test correction given the size of the gene set (e.g., false discovery rate correction)? Was this experiment analyzed as a fully specified statistical model, or were individual treatments compared to individual control treatments separately from others, and if so was there multiple test correction for that? There is insufficient information provided here to be able to evaluate the quality and robustness of the gene expression analysis. Given that gene expression analysis is a focus of the paper this is a problem.

41.9 million reads (subsection “Extraction of mRNA, RNA sequencing and bioinformatics”) – is this per sample?

32.69% mapping efficiency (subsection “Extraction of mRNA, RNA sequencing and bioinformatics”) – this is extremely low! Why is this so low if sequence similarity between haddock and reference (cod) genome is so high? How was successful mapping assessed – how were mapping parameters set? This should be a red flag that something went wrong in the mapping, or that there was significant contamination issues.

After mapping, this left 31.51 million reads "for each group". I'm not sure what this means "for each group". It is generally accepted that a minimum of ~10M reads PER SAMPLE (that is, per experimental unit) is adequate for RNA-seq.

"three biological replicates per stage” – section "Total RNA and cDNA preparation" states that RNA was from pools of embryos. So are these more accurately 3 replicate pools?

KEGG analysis: what were the criteria used to include a KEGG pathway in figures S4/S5? At least one of the criteria should be statistically significant enrichment. From my look at these figures, there are no obvious pathway-level connections to the phenotypes/physiological pathways on which the authors hang their hat. E.g., these figures do not have any terms that obviously relate to cholesterol homeostasis. Nor does Figure 3—figure supplement 1 indicate any enrichment of pathways having to do with ionoregulation, or craniofacial development. So what is the analysis here?

Selected gene sets: The authors make assertions about physiological mechanisms given responses of specific genes with inferred functions that make a nice story. However it is unclear what is the null expectation from this type of analysis.

E.g. Subsection “Genes associated with defects in cardiac function and morphogenesis”: "We therefore focused on genes associated with cardiomyocyte membrane potential and intracellular Ca^2+^ cycling" – how were these genes selected?

In the same section: "The expression of a few E-C coupling genes" – was there an unbiased collection of E-C coupling genes, then test for significant enrichment of these genes?

In the same section: "genes involved in cardiac morphogenesis" – again, how were these collected/curated as a gene set? Please provide the query gene set.

Subsection “Genes associated with craniofacial abnormalities”: "We therefore interpreted developmental changes in haddock gene expression in the context of these well-characterised zebrafish mutants." – so the "tester" set here was all genes shown in zebrafish mutants to be involved in neural crest cell or muscle development? Please include this set of genes. The 12 genes indicated below this section – is this a significant enrichment?

Subsection “Genes associated with ion and water regulatory imbalance”: "Our analysis focused on key ionoregulatory proteins in MRCs and their associated genes, including Na+/K^+^ ATPase subunits (*at1* genes, e.g. *at1a1-a4, at1b1-b4*), a urea transporter (*ut1*), a Na+/K^+^/2Cl^-^ co-transporter (*s12a2*), the sodium hydrogen exchanger Nhe3 (*slc9a3*), and a chloride channel" – this is clearly not a complete or objective curation of ionoregulatory genes.

Is there an unbiased query gene set for any of these analyses? This whole "genes associated with phenotypes" section – is there objective analysis? What makes this anything more than just story telling given some cherry-picked genes? The authors should at least use language that proposes some hypotheses, rather than writing text so as to imply strong conclusions of cause-effect adverse outcome pathways. This is a problem that permeates the reporting of results and discussion, and contributes in a large way to overstatement of the results that are expanded on below.

Unsupported claims and overstatement issues:

There are pervasive issues throughout the manuscript, often associated with claims or conclusions that are (or should be) in fact stated as hypotheses. It is often unclear whether certain statements are supported by their data, supported by the literature, or are speculation or informed hypotheses. Some examples follow:

Subsection “Genes associated with defects in cardiac function and morphogenesis”: "Overall, the transcriptional response to disrupted E-C coupling was not a simple pattern of compensation" – how do they know that this is a transcriptional response to E-C coupling? Is expression not being measured in whole animals? What proportion of overall tissue mass is accounted for by the heart? I think the authors are drawing conclusions where they ought to be proposing hypotheses. This is an issue in MANY places throughout the manuscript

Subsection “Genes associated with craniofacial abnormalities”: "Overall, the genes identified above for both neural crest and muscle lineages represent a more complex pattern of dysregulation than previously been reported for any of the individual zebrafish craniofacial mutants. This suggests that crude oil is acting on novel developmental processes." – It is unclear what is supporting this assertion (1st sentence), and how the assertion leads to the conclusion (2nd sentence)

Subsection “Genes associated with ion and water regulatory imbalance”: "and they should therefore lose water along a diffusion gradient if osmoregulation is disrupted as a consequence of heart and circulatory failure." – Does oil exposure cause ionoregulatory dysfunction and/or water loss in developing marine fish? Are these statements speculation/hypotheses, or are they supported in the literature? There are several prominent studies on the effects of oil exposure on osmoregulation in fish, but none of these are referred to.

In the same section: "a third distinct oil-induced adverse outcome pathway." – how do they know that this is a pathway distinct from cardiac dysfunction? E.g., rather than just a secondary manifestation of circulatory failure?

"Crude oil exposures therefore appear to cause osmotic stress in the developing embryonic nervous system" – the authors are implying an overall impact on ionoregulatory abilities? Is there any support for this here or in the literature? E.g., any studies showing dysfunction of net sodium or chloride flux upon oil exposure? Do the authors mean to propose this as a hypothesis?

Subsection: "A novel adverse outcome pathway: disruption of cholesterol homeostasis" – this subheading, and the following paragraphs, claim novelty, and appear to claim precedent for this discovery. Has oil exposure impacts on yolk utilization not previously been reported in the literature? I can find a publication from the 1990s after just a couple minutes searching. Furthermore, the studies presented here do not provide direct evidence for impacts on yolk mobilization or cholesterol homeostasis. E.g., how do the authors know that the apparently increased yolk size in Figure 1 isn't just a result of fluid accumulation stemming from observed edema?

In the same section: "Pathway analysis was also consistent with a significant effect on cholesterol homeostasis” – The associated figure is is missing. Also, staring at Figure 3—figure supplement 1, I see no pathways that include any terms obviously related to cholesterol homeostasis. Nor for that matter does Figure 3—figure supplement 1 indicate any enrichment of pathways having to do with ionoregulation, or craniofacial development,

Subsection “Unaltered gene expression in relation to visibly normal organs: lateral line and liver.”: "Similarly, the related KEGG pathways that are inclusive of these genes were relatively unaffected by oil exposure at all time points" – but what does this really mean? KEGG pathways (Figure 3—figure supplement 1 and Figure 3—figure supplement 2), as far as I can tell, don't implicate many (any?) of the mechanisms that the authors are building their story on – e.g., ionoregulation, cholesterol homeostasis, craniofacial development. If this paragraph was intended to serve as a test or confirmation that their conclusions/assertions in preceding paragraphs have merit, then this is unconvincing – and problematic for supporting the assertions that the other functions (ionoregulation, cholesterol homeostasis, craniofacial development) ARE supported by pathway-level analysis.

Discussion section:

"We identified specific changes in the expression of key genes involved in the function or morphogenesis of individual tissues and organs with visible abnormalities. Given unaltered gene expression associated with apparently unaffected structures such as the liver, the DEGs in oil-exposed haddock indicate a disruption of specific developmental processes, as opposed to non-specific effects (e.g., general developmental delay)." – this appears to me as a gross overstatement. There was no organ-specific analysis of gene expression. The authors measured gene expression in whole animals. They therefore are not in a position to make assertions about expression in the heart or liver, though of course they are free to propose hypotheses.

"Our data demonstrate a transcriptional cascade that is tightly linked to these defects in cardiac function (cardiomyocyte intracellular calcium cycling) and form (heart chamber growth)." – there are 4 genes that are slightly down-regulated at one timepoint preceding the emergence of altered cardiac phenotypes (6 dpf), and none deferentially expressed immediately post-organogenesis (10 dpf) when the proposed E-C uncoupling should be apparent. Does this represent the discovery/demonstration of a "transcriptional cascade"?

"Crude oil likely disrupts normal MRC function" – what is the evidence for this?

"MRC function could be impaired by a high metabolic cost of PAH degradation." – why? Evidence or rationale to support this?

"In haddock embryos, fluid moves from the dorsal subdermal space to the yolk sac. At the larval stage, the permeable yolk sac membrane is replaced by the more resistant peritoneal cavity and body wall, causing fluid to move into the dorsal finfold and adjacent tissues." – I assume that the authors mean to write this as a proposal or hypothesis that is consistent with their observations?

Discussion section paragraph nine: much of this is highly speculative. There are plenty of transcriptomics studies of oil exposure during fish development, that include edema as an endpoint. If the authors want to make these assertions perhaps they should check those other studies for altered regulation of VEGF-C.

Discussion paragraph ten: This entire paragraph reads as excessive speculation

Discussion paragraph eleven: "but tissue localization during craniofacial development in embryonic fish is needed to confirm a role in this adverse outcome pathway" – look up Planchart & Mattingly 2010 TCDD upregulates FOXQ1 in zebrafish jaw primordium

Discussion section: "First, we used known spatial mRNA distributions in model species (primarily zebrafish) to more accurately phenotypically anchor the transcriptome data for crude oil-exposed haddock" – this sounds fancy but there is no description of this in the methods

Reviewer #2:

In their manuscript "Novel adverse outcome pathways revealed by chemical genetics in a developing marine fish", Sorhus et al. characterize the impact on the transcriptome of PAHs during embryonic/larval development of Atlantic haddock. The authors examined gene expression profiles from fish exposed using three different exposure paradigms, two chronic (low =.58 µg/L & high = 6.7 µg/L) and one intermittent (6.1 µg/L per pulse). The authors examine genes that underlie one of four phenotypes observed in exposed embryos/larvae and show that changes in these genes may lead to the phenotypes observed. The authors' results will be of interest to the readership of *eLife*. However, several concerns exclude publication of the manuscript in its current form.

Major concerns:

1) Given that haddock is not a model system, better explanation of the developmental time windows will be critical for most readers to understanding the context of the embryonic and larval developmental stages. In zebrafish 50% epiboly occurs at 5.5 hpf while in haddock it occurs at 3 dpf.

2) Authors need to add Alcian images of the new facial phenotypes not listed in their previous work, the upper jaw is not shown in previously nor is the basicranium.

3) The authors need to be more explicit in their logic in moving from a broad overview of the transcriptome and to genes that directly impact the phenotypes listed when those genes are not the most highly responsive. Why discuss the broader transcriptional changes (subsection “General patterns of gene regulation in response to crude oil”)?

4) Without the ability to create genetic chimeras or cell-type specific transgenics to directly test the tissue target of PAHs effect on facial development, the authors need to soften their stance that PAH disrupts muscle development thereby affecting skeletal development. Crump and Schilling have shown that in zebrafish *edn1* is expressed in and required in NCCs. The authors cannot rule out a direct NCC-PAH impact based only on their data (Discussion section).

5) What do the authors mean by the "transcriptional response to disrupted E-C coupling was not a simple pattern of compensation" (subsection “Genes associated with defects in cardiac function and morphogenesis”)?

6) Several the figures need to be reviewed. Two figure supplements are missing from the manuscript, though they are described in the text (subsection “A novel adverse outcome pathway: disruption of cholesterol homeostasis”). Figure 6 shows that myh1 is down regulated at 0 dph not 11 dpf as described in the text (subsection “Genes associated with craniofacial abnormalities”). The supporting datasets need to be organized with the supplemental Table 1 for readability. It is difficult to relate these datasets with the table in the current form.

Reviewer #3:

Overall assessment: This is an interesting study that is well done overall and advances our understanding of oil impacts on developing marine fish. Strengths include the use of an environmental relevant non-model organism, the experimental design involving exposure at both embryonic and larval stages, use of multiple modes of exposure (two concentrations and a pulsed exposure), sampling at multiple time points, and a rich set of data on gene expression that is anchored to phenotypes.

Substantive concerns:

1) In the title, Abstract, and manuscript, the authors overreach when they invoke "chemical genetics" and "adverse outcome pathways" (AOPs). Chemical genetics involves high-throughput screening of libraries of individual compounds, a much more precise approach than the exposure to a complex chemical mixture performed here. Although nowhere defined by the authors, an AOP describes the entire sequence of events from a molecular initiating event, across multiple levels of biological organization, to an adverse outcome; it is synonymous with "mechanism of action" (see e.g. Ankley 2010 Envir. Tox. Chem. and Villeneuve 2011 Envir. Tox. Chem.). When the authors refer to AOPs they are actually referring only to adverse outcomes, not the pathways. Their gene expression data may help to inform our understanding of some AOPs associated with oil exposure, but they certainly have not "revealed novel AOPs." And contrary to the claim in the Abstract, it is not clear that they have "identified initiating events"-the specific chemical-protein interactions that lead to the gene expression and phenotypic changes that they report.

2) The authors' approach, which is stated explicitly (Results section), is to interpret changes in haddock gene expression in relation to known zebrafish mutants, i.e. they focused only on specific genes known to be involved in development of the tissues affected by oil. While this is valuable, is there a more objective approach that might be used to identify unexpected associations between changes in gene expression and specific phenotypes? It seems as though they have not taken full advantage of the unbiased RNA-seq dataset in this regard.

3) The authors over-interpret the connections between gene expression patterns and phenotypes, claiming cause-effect relationships in haddock from what are only associations. For example, Discussion section paragraph two claims a "tight linkage" between gene expression and cardiac defects. Whether the gene expression changes are causal or are secondary to the phenotypic changes is not clear. The authors could strengthen their arguments by being more explicit about the temporal and concentration-dependent associations between gene expression patterns and phenotypes, e.g. by adding a measure of phenotypic progression to Figure 4–Figure 7.

4) The experimental design is not described sufficiently. The methods (paragraph one) refer to a previous paper, but it is not clear whether these samples are from the same experiment described in that paper, or just used similar exposures. Indeed, the oil concentrations in that paper are expressed differently than in the current manuscript. Even if the methods are the same, this paper should be a description of the experimental design, including exposure conditions, numbers of replicates, numbers of pooled embryos in each replicate, etc.

[Editors' note: further revisions were requested prior to acceptance, as described below.]

Thank you for submitting your article "Novel adverse outcome pathways revealed by chemical genetics in a developing marine fish" for consideration by *eLife*. Your article has been reviewed by three peer reviewers, and the evaluation has been overseen by a Reviewing Editor and Marianne Bronner as the Senior Editor. The reviewers have opted to remain anonymous.

The reviewers are generally happy with your revisions but request some clarifications in methodology and also raise some minor comments. I ask you to correct these points prior to passing on your manuscript for production. Individual reviews are below.

Reviewer #1:

Details of the experimental design and analysis are still not adequate. The authors refer the reader to the Sorhus 2016b publication for all aspects of experimental design. The authors should provide in the current manuscript at least the basics of experimental design in the methods section, including the number of doses, the specific developmental stages examined within each dose, and the number of biological replicates (replicate pools) within each dose*stage treatment. Furthermore, I have read Sorhus 2016b and it is not obvious WHAT were the exact contrasts without much effort on the part of the reader. The authors should make it easy to understand the basic of the experimental design.

Also, it is still not clear to me what was the unit of replication. I asked for clarification on the nature of replication previously. The revised manuscript reads "cDNA library preparation and sequencing was performed by the Norwegian Sequencing Centre (NSC, Oslo, Norway) on one pool from three replicate tanks per stage for each dose (low, pulse and high) plus control using the Illumina TruSeq RNA Sample Preparation Kit".

A treatment is a dose*stage. Were there replicate pools of embryos assayed per treatment? As far as I can tell, six developmental stages were profiled at each of 3 doses and a control, resulting in 24 treatments. I find 24 "samples" sequenced in the SRA (https://www.ncbi.nlm.nih.gov/Traces/study/?acc=SRP060012). This would suggest one pool per treatment, unless there are multiple samples embedded within each SRA entry. If there is just one sequenced pool per dose*stage treatment, then I recommend rejection of the manuscript for lack of replication of experimental units. If I misunderstand this, and there is in fact replication of pools of embryos within each dose*stage treatment, then I recommend the following:

Subsection “Structure of pelagic larvae and visible phenotypes associated with crude oil exposure”: "with 96% of high dose animals showing abnormal phenotypes, ranging to ~ 60% for pulse dose and ~ 35% for the low dose."

The authors should consider preparing a visual that summarizes this distribution of phenotypes across doses. E.g., stacked bar chart (or pie chart), including proportion of animals showing each phenotype, with dose as series. And perhaps separate plots for different developmental stages.

Subsection “Oil-induced changes in gene expression during embryonic development.”: "p > 0.05" Is it not more standard to notate as p<0.05?

Subsection “General patterns of gene expression in response to crude oil”: "At all stages, the subcategory of Organismal Development or Embryonic Development (henceforth combined as Development) was in the top 5 Diseases and Bio Functions category under Physiological System Development and Function with p values ranging from 10-3 to 10-19 " Are categories/subcategories of functions from IPA? Authors should state this.

In the same subsection: "Pathway enrichment was dose-dependent and clearly associated with the frequencies of abnormal phenotypes ([Supplementary-material SD1-data])." It would appear that reviewers do not have access to these supplementary files. This is frustrating.

Subsection “Genes associated with defects in cardiac function and morphogenesis”: "overexpression of *bmp10, nkx25*, or *tbx3* is associated with serious heart defects in other vertebrates." This needs a citation

Discussion section: "Two major initiating events for crude oil-associated cardiac defects during fish development are chemical blockade of IK_r_ repolarizing potassium currents, (encoded by *kcnh2*) and disruption of intracellular calcium handling, the latter culminating in sarcoplasmic reticulum (SR) calcium depletion through effects on either RyR or SERCA2 (encoded by *ryr2* and *at2a2*, respectively). In the fully formed heart, these pharmacologic effects impair cardiac function by inducing arrhythmia and reducing contractility." Citations are needed in this section.

Subsection “Extraction of mRNA, RNA sequencing and bioinformatics”: "150 key genes involved in cardiac and craniofacial development and cardiac function were assembled." Please provide a table of these genes, and relevant citations describing their relationships to key phenotypes. The authors state in their rebuttal "The lists are provided in a new table, [Supplementary-material SD1-data]." It would appear that I unfortunately do not have access to review these files (or I'm somehow just looking in the wrong place?). If not already done, authors please make sure these files are detailed with the phenotype relationship and citations to all relevant literature (e.g., defend criteria for including a gene in your curated set).

In the same subsection: "The contigs" is this the reference cod sequence, or the haddock RNA-seq read sequences?

The authors state in their rebuttal "Although the effects of oil exposure on salt and water balance in fish embryos have not been examined," This is perhaps true, but it has been examined in adults, and this may be worth noting in the manuscript. E.g.: Kennedy CJ, Farrell AP (2005) Ion homeostasis and interrenal stress responses in juvenile Pacific herring, *Clupea pallasi*, exposed to the water-soluble fraction of crude oil. Journal of Experimental Marine Biology and Ecology 323, 43-56.

Regarding the author's rebuttal "As for direct evidence of disrupted cholesterol homeostasis, we are not sure what is more direct than up-regulation of HMG-CoA-reductase" Up regulation of a gene is not a direct measure of altered cholesterol homeostasis.

Reviewer #2:

In their revised manuscript "Novel adverse outcome pathways revealed by chemical genetics in a developing marine fish", Sorhus et al. characterize the impact on the transcriptome of PAHs during embryonic/larval development of Atlantic haddock. The authors examined gene expression profiles from fish exposed using three different exposure paradigms, two chronic (low =.58 µg/L & high = 6.7 µg/L) and one intermittent (6.1 µg/L per pulse). The authors examine genes that underlie one of four phenotypes observed in exposed embryos/larvae and show that changes in these genes may lead to the phenotypes observed. The authors have addressed concerned all concerns raised, therefore the results in the revised manuscript will be of interest to the readership of *eLife* and is ready for publishing in its current form.

Reviewer #3:

I would quibble with a few of the responses to my original comments, but I don't view these issues as serious enough to derail publication of the paper. Nevertheless, I point them out for the authors' consideration.

1) Regarding what is an "initiating event" in an adverse outcome pathway (AOP), I disagree with the authors' claim (response to reviewer #3, point 1) that *bmp10* upregulation is such an initiating event. It is an early event, certainly. But the true initiating event is the chemical-protein interaction that causes *bmp10* to be upregulated. That event has not been identified in this paper.

2) With regard to my questioning of their statement claiming that it has not been clear that oil influences mRNA levels as part of a developmental phenotype (Discussion section): The authors pointed out that the several previous papers I noted as showing oil affecting mRNA expression in fish did not involve measurements made in embryos. However, they did not provide this explanation in the revised manuscript, despite the fact that two of the three reviewers raised the same question. In addition, in their response the authors mention a more recent paper, published (27 June) prior to the original submission of this manuscript (30 Aug), that does include transcriptomic analysis of fish embryos exposed to oil (Xu et al. ES&T). It is unfortunate that the authors did not take advantage of the opportunity to better explain their original statement or compare their results with the prior work in embryos.

3) The manuscript would be improved by a clearer summary of the experimental design so that the readers don't have to dig out the other paper. And the point about replication is important – this needs clarification.

[Editors' note: further revisions were requested prior to acceptance, as described below.]

Thank you for resubmitting your work entitled "Novel adverse outcome pathways revealed by chemical genetics in a developing marine fish" for further consideration at *eLife*. Your revised article has been favorably evaluated by Marianne Bronner as the Senior editor, a Reviewing editor, and one reviewer.

The manuscript is in principle ready for publication pending a minor but critical revision which is to include relevant information rather than referring the readers to other papers. The reviewers have asked for this repeatedly and I am only prepared to accept your paper with the inclusion of this additional information. This is an easy change and I hope you will make it quickly. Below is the comment from the reviewer.

Reviewer #1:

The revisions, in general are fine. However, I should note that I have been generally frustrated by the struggle with the authors to include all relevant information, accessibly, within this manuscript. Far too many papers are published these days that have incomplete description of methods. Also, many papers refer the reader to other papers to find out details of the methods. This results in un-necessary additional work for the reader – especially un-necessary since all journals these days have supplemental sections that allow authors to include all relevant information, without cluttering up the main manuscript. The authors are still insisting on sending the review on a hunt through their previous papers to find relevant information ("A thorough functional description of the genes including citations are provided in Sørhus et al. 2016a."). This is not too much to ask. We should all strive to make our research more transparent, and more reproducible.

---

## [Author Response]

*As you will see from the individual reviews below, all of the reviewers had extensive issues with the analysis and lack of detail in the description of method. Even more importantly, however, they also felt that the there was a great deal of over-interpretation and "overselling of the work", rendering many statements unsupported by the data. We feel that this is fixable but will require considerable rewriting and re-review. If you feel you are able to revise the paper accordingly, we would be willing to consider a significantly revised version for re-review.*

We appreciate the careful reviews and constructive comments from each of the three referees. We would like to initially provide a response that addresses many of the larger concerns of reviewers 1 & 3. Both helpfully pointed out that we failed to provide sufficient detail on our bioinformatics approach. Our original submission contained only a very brief description of our approach – i.e., “Individual genes involved in cardiac and craniofacial development, osmoregulation and lipid metabolism, or not directly linked to KEGG pathways were curated through manual searches of the literature (via PubMed and Web of Science) and expression databases at the Zebrafish Information Network (www.zfin.org).” We have explained our methods in much more detail in this revised submission.

Unlike conventional toxicogenomics, which relies primarily on bioinformatics to identify response pathways, our approach was more based on developmental genetics. We felt this was necessary because of the uncertainty inherent in cross-species extrapolations from a commercial ‘omics platform (e.g., designed for mammals) to a non-model fish (Atlantic haddock). In a genetic screen, investigators typically select the most robust (e.g., fully penetrant) phenotypes for in-depth analyses. In the original manuscript we focused on results from the high dose treatment group, because virtually 100% of the animals showed the abnormal phenotypes. This allowed for robust linkages between specific phenotypes and gene expression, and also kept the paper to a manageable length.

Prior to the current study, we characterized the normal developmental transcriptome of haddock (Sørhus et al., 2016; Dev. Biol. 411:301). This necessitated an extensive evaluation of the literature for vertebrate cardiac and craniofacial development to identify the most important genes for heart development/function as well as craniofacial/bone development (~ 150 genes for each category). The basic read count data were then used to identify all DEGs, and initial software-based statistical bioinformatics was used to compare changing patterns of gene expression at different developmental time points. Profiling the normal transcriptome was very labor-intensive; two members of our team worked on the project for the better part of a year.

This earlier gene identification and prioritization effort served as a cornerstone of our current study. This approach yielded very clear linkages between critical genes for which a loss or gain of function by mutation led to phenotypes resembling those observed in oil-exposed haddock. Moreover, we made novel observations of transcriptional regulation for clusters of functionally related genes, such as those involved in cholesterol synthesis and transport. This is a major new insight into developmental oil toxicity in fish, and will lead to new metrics for measuring injury at both the gross morphological (e.g., yolk absorption) and molecular scales.

The above explanation aside, we appreciate the reviewers’ comments related to a more conventional toxicogenomic analysis. Accordingly, during the revision phase, we have reanalyzed our data using Qiagen’s Ingenuity Pathway Analysis (IPA). We are pleased to report that each of the relationships identified in our original submission has been confirmed with statistical rigor. Therefore, our manual curation of the haddock transcriptome reinforces our confidence in an informatics application such as IPA to detect statistical enrichments of causal injury response pathways. This is uncommon, and we believe the paper is therefore strengthened. We also reiterate the importance of manual curation, as IPA also identified pathways that were misleading and apparently based on very tenuous associations in the literature.

*Reviewer #1:*

*In this article the authors expand on previous experiments, where developing haddock are exposed to low levels of crude oil, to test for impacts on sensitive developmental endpoints. Developmental impacts are described in detail in previous reports, and this study reports transcriptomic changes during development following three oil exposure regimes.*

*I have two main difficulties with this paper. First, there is insufficient description of methods for gene expression analyses. Second, there is much speculation and unsupported statements such that much of the manuscript reads as over-statement.*

*Gene expression analysis:*

*Though the experimental design includes three different exposure regimes, specific gene transcriptional responses are not correlated in any way to dosing regime (high, low, pulse). The authors ought to show gene expression dose responses, at least for the genes for which they are trying to build cause-effect relationships with adverse outcomes. More convincing arguments for cause-effect would be supported by data showing that dysregulation of the gene precedes the appearance of the phenotype, and that dysregulation is more pronounced in treatments with more pronounced phenotypic effects (e.g., higher doses). The lack of detail reported here could make a reader concerned that perhaps interpretation of gene-specific responses are not as clear-cut as represented in the text. Are Figure 4, Figure 5, Figure 6, Figure 7 showing data from any dose, or just high dose? Furthermore, there is no description of how emergence of phenotypes varied with dosing regime. Text (starting paragraph two of the Results section) that describes developmental phenotypes does not distinguish between effects that differ between doses, and all of the images from exposed animals in Figure 1 appear to be only from the high dose.*

As noted above, we focused on the high dose treatment group because virtually 100% of the animals had the full range of abnormal phenotypes. This was clarified in the figures. However, as the reviewer suggests, we may not have provided a sufficiently detailed description of each phenotype – this was published previously (Sørhus et al., 2016; Sci. Rep. 6:31058). To address this, we have revised Figure 2 to more clearly relate changes in gene expression to the developmental onset of each phenotype. We also provided more information on the dose-dependency of gene expression in a new table ([Supplementary-material SD1-data]), including the top IPA-identified pathways (e.g., Molecular and Cellular Function, etc.) for each dose and developmental time point.

*Other aspects of gene expression analysis that are problematic:*

*There is no description of the statistical model. What was compared to what? How many tests were there? Was there multiple test correction given the size of the gene set (e.g., false discovery rate correction)? Was this experiment analyzed as a fully specified statistical model, or were individual treatments compared to individual control treatments separately from others, and if so was there multiple test correction for that? There is insufficient information provided here to be able to evaluate the quality and robustness of the gene expression analysis. Given that gene expression analysis is a focus of the paper this is a problem.*

Each treatment (oil-exposed or unexposed controls) consisted of three pools.

We have used NOISeqBIO for the RNA-seq analysis where the 3 pools per treatment per lifestage analysed were treated as replicates. NOISeqBIO returns a DE (differential expression) probability that is equivalent to FDR (False discovery rate) adjusted P-values. The following reference has been added in subsection “Real time qPCR” to make this more clear:

Tarazona S, Furio-Tari P, Turra D, Pietro AD, Nueda MJ, Ferrer A, Conesa A (2015) Data quality aware analysis of differential expression in RNA-seq with NOISeq R/Bioc package. Nucleic Acids Research 43:e140.

*41.9 million reads (subsection “Extraction of mRNA, RNA sequencing and bioinformatics”) – is this per sample?*

Yes. To clarify, we changed the following sentence:

“The total number of reads averaged 41.39 million per sample and the mapping efficiency averaged 32.69%, giving an overall average of 13.51 million mapped paired end reads (125 bp) for each sample.”

*32.69% mapping efficiency (subsection “Extraction of mRNA, RNA sequencing and bioinformatics”) – this is extremely low! Why is this so low if sequence similarity between haddock and reference (cod) genome is so high? How was successful mapping assessed – how were mapping parameters set? This should be a red flag that something went wrong in the mapping, or that there was significant contamination issues.*

We mapped the haddock data to Atlantic cod because the genome of the latter is well annotated and nearly complete. The cod genes are listed with ensemble-IDs, allowing other investigators to extract the sequences to which we’ve mapped our haddock sequences. Also, the cod genome has served as a reference in several previous projects. Of 20613 sequences, 19000 cod genes map over 10 reads to haddock in at least one sample. We agree that 32% mapping efficiency appears low, but this is unlikely to reflect missing genes or a contamination issue. Rather, we strictly counted the reads that mapped exclusively to exons. Untranslated regions and alternatively spliced transcripts were not included. Also, the cod genome model is relatively fragmented, to the extent that RNA-seq in cod yields approximately 40% mapping to the exons of the model (Star et. al, 2011; Nature 477:207-210). Lastly, we expect some sequence variation across the two species.

*After mapping, this left 31.51 million reads "for each group". I'm not sure what this means "for each group". It is generally accepted that a minimum of ~10M reads PER SAMPLE (that is, per experimental unit) is adequate for RNA-seq.*

The number of reads per sample was high – i.e., the lowest number of reads for any sample after mapping was more than 10 M reads.

*"three biological replicates per stage” – section "Total RNA and cDNA preparation" states that RNA was from pools of embryos. So are these more accurately 3 replicate pools?*

This has been clarified in the Methods as follows: “cDNA library preparation and sequencing was performed by the Norwegian Sequencing Centre (NSC, Oslo, Norway) on one pool from three replicate tanks per stage for each dose (low, pulse and high) plus control using the Illumina TruSeq RNA Sample Preparation Kit”.

*KEGG analysis: what were the criteria used to include a KEGG pathway in figures S4/S5? At least one of the criteria should be statistically significant enrichment. From my look at these figures, there are no obvious pathway-level connections to the phenotypes/physiological pathways on which the authors hang their hat. E.g., these figures do not have any terms that obviously relate to cholesterol homeostasis. Nor does Figure 3—figure supplement 1 indicate any enrichment of pathways having to do with ionoregulation, or craniofacial development. So what is the analysis here?*

Please see our overarching response to this issue above (Editor’s comments). The additional analyses we’ve performed using IPA validates our manual curation (and vice versa), and also addresses the issue of statistical enrichment.

*Selected gene sets: The authors make assertions about physiological mechanisms given responses of specific genes with inferred functions that make a nice story. However it is unclear what is the null expectation from this type of analysis.*

*E.g. Subsection “Genes associated with defects in cardiac function and morphogenesis”: "We therefore focused on genes associated with cardiomyocyte membrane potential and intracellular Ca^2+^ cycling" – how were these genes selected?*

*In the same section: "The expression of a few E-C coupling genes" – was there an unbiased collection of E-C coupling genes, then test for significant enrichment of these genes?*

*In the same section: "genes involved in cardiac morphogenesis" – again, how were these collected/curated as a gene set? Please provide the query gene set.*

It seems more appropriate to consider null hypotheses in relation to specific structures or specific signaling pathways. For example, the hypothesis that genes governing phenotypically altered and unaltered structures would be equally responsive to crude oil. As stated in the subsection entitled “Unaltered gene expression in relation to visibly normal organs: lateral line and liver”, we essentially rejected this null hypothesis. For signaling pathways, a null hypothesis might be that Hedgehog signaling and Hox genes are affected in ways that are similar to BMP signaling. Our data do not support this hypothesis because developing haddock do not have phenotypes associated with disrupted Hedgehog or Hox signaling, and we observed minimal to no change in the transcription of these genes. As noted above, our priority gene lists were derived from an extensive evaluation of the developmental biology literature, and have since been confirmed through rigorous statistical analysis. The lists are provided in a new table, [Supplementary-material SD1-data].

*Subsection “Genes associated with craniofacial abnormalities”: "We therefore interpreted developmental changes in haddock gene expression in the context of these well-characterised zebrafish mutants." – so the "tester" set here was all genes shown in zebrafish mutants to be involved in neural crest cell or muscle development? Please include this set of genes. The 12 genes indicated below this section – is this a significant enrichment?*

The gene set is now provided in [Supplementary-material SD1-data].

*Subsection “Genes associated with ion and water regulatory imbalance”: "Our analysis focused on key ionoregulatory proteins in MRCs and their associated genes, including Na+/K^+^ ATPase subunits (at1 genes, e.g. at1a1-a4, at1b1-b4), a urea transporter (ut1), a Na+/K^+^/2Cl^-^ co-transporter (s12a2), the sodium hydrogen exchanger Nhe3 (slc9a3), and a chloride channel" – this is clearly not a complete or objective curation of ionoregulatory genes.*

Is there an unbiased query gene set for any of these analyses? This whole "genes associated with phenotypes" section – is there objective analysis? What makes this anything more than just story telling given some cherry-picked genes? The authors should at least use language that proposes some hypotheses, rather than writing text so as to imply strong conclusions of cause-effect adverse outcome pathways. This is a problem that permeates the reporting of results and discussion, and contributes in a large way to overstatement of the results that are expanded on below.

For simplicity, we provided data on a small list of the most important actors in ionocyte function, as determined from a current review of the literature on iono- and osmoregulation in fish. The revised manuscript now provides a secondary analysis in IPA, statistically confirming alterations in a broader set of genes involved in ion transport.

*Unsupported claims and overstatement issues:*

*There are pervasive issues throughout the manuscript, often associated with claims or conclusions that are (or should be) in fact stated as hypotheses. It is often unclear whether certain statements are supported by their data, supported by the literature, or are speculation or informed hypotheses. Some examples follow:*

*Subsection “Genes associated with defects in cardiac function and morphogenesis”: "Overall, the transcriptional response to disrupted E-C coupling was not a simple pattern of compensation" – how do they know that this is a transcriptional response to E-C coupling? Is expression not being measured in whole animals? What proportion of overall tissue mass is accounted for by the heart? I think the authors are drawing conclusions where they ought to be proposing hypotheses. This is an issue in MANY places throughout the manuscript*

The role of transcription in the control of E-C coupling was discussed in detail in a previous paper that fully describes all aspects of the cardiac phenotypes (Sørhus et al., 2016; Sci. Rep. 6:31058). For brevity, we did not repeat that in-depth description in our original submission. However, it is clear from the reviewer’s comment that more clarification is needed, and we have revised the text in the “Genes associated with defects in cardiac function and morphogenesis” section accordingly. The current study specifically tested a hypothesis concerning transcriptional responses to disrupted E-C coupling. Very few studies have measured the responses of E-C coupling genes (i.e., ion channels) to pharmacologic blockade – these are cited in the original and revised version of our manuscript. Based on this limited information, we hypothesized a compensatory up-regulation of either the cognate gene for the blocked channel or other channel genes that function in the same pathway. This predicts, for example, an upregulation of the kcnh2 gene in response to potassium channel blockade. Because we observed the opposite, and did not see a compensatory upregulation of genes controlling repolarization, we concluded that the overall response was not a simple pattern of compensation. This suggests that crude oil is having a dual effect on E-C coupling at both the protein and gene levels.

In terms tissue specificity, as discussed in responses above, we assessed many genes that are expressed solely in the heart at a given developmental stage. Our RNA-Seq data demonstrate that these genes can be quantified despite the very contribution of the heart to total embryo mass. To address issues of tissue specificity, contribution of different tissues to the mass of embryos and larvae, and read count data, we added a new section and a table ([Supplementary-material SD1-data]) to the Results.

*Subsection “Genes associated with craniofacial abnormalities”: "Overall, the genes identified above for both neural crest and muscle lineages represent a more complex pattern of dysregulation than previously been reported for any of the individual zebrafish craniofacial mutants. This suggests that crude oil is acting on novel developmental processes." – It is unclear what is supporting this assertion (1st sentence), and how the assertion leads to the conclusion (2nd sentence)*

Again, we wanted to keep this simple for brevity. There are some predictions about gene relationships based on the analysis of zebrafish mutants in terms of up- or down-regulation of signaling molecules, receptors, and downstream effectors. As is the case for E-C coupling genes, the pattern does not match the predictions based on single gene mutations in zebrafish. The oil-induced phenotype and the pattern of gene dysregulation are not a clear phenocopy of any one zebrafish craniofacial mutant. We have added text at this point in the manuscript to clarify.

*Subsection “Genes associated with ion and water regulatory imbalance”: "and they should therefore lose water along a diffusion gradient if osmoregulation is disrupted as a consequence of heart and circulatory failure." – Does oil exposure cause ionoregulatory dysfunction and/or water loss in developing marine fish? Are these statements speculation/hypotheses, or are they supported in the literature? There are several prominent studies on the effects of oil exposure on osmoregulation in fish, but none of these are referred to.*

We provided the justification for this statement in the original Discussion based on studies in saltwater medaka. Although the effects of oil exposure on salt and water balance in fish embryos have not been examined, there is a known role for cardiac circulation. Based on knockdown studies that disrupt circulation by causing heart defects, marine embryos can be expected to lose water and gain salt, with total osmolality increasing. We therefore infer that fluid accumulation in the vicinity of the yolk sac (edema) must represent free water moving from other compartments within the embryo. The visible phenotype is consistent with this – i.e., a reduction of the marginal subdermal space and a loss of cerebrospinal fluid (hindbrain ventricle). We also suggest several additional hypotheses regarding other mechanisms by which crude oil could disrupt ionoregulation more directly.

*In the same section: "a third distinct oil-induced adverse outcome pathway." – how do they know that this is a pathway distinct from cardiac dysfunction? E.g., rather than just a secondary manifestation of circulatory failure?*

We are drawing a distinction from the canonical cardiotoxicity pathways relating to sublethal heart malformation. Disturbances of salt and water balance, whether they are indirectly due to circulatory disruption or a consequence of direct PAH toxicity to ionocytes, are likely to represent adverse outcomes that have been previously unrecognized in this field, over three decades of research. For example, transcriptional changes observed in CNS-specific osmoregulatory genes may reflect a dysregulation of CNS development. Importantly, we also see changes in ion transport genes relatively early in development, prior to visible cardiac dysfunction. We have clarified this in the revised text.

*"Crude oil exposures therefore appear to cause osmotic stress in the developing embryonic nervous system" – the authors are implying an overall impact on ionoregulatory abilities? Is there any support for this here or in the literature? E.g., any studies showing dysfunction of net sodium or chloride flux upon oil exposure? Do the authors mean to propose this as a hypothesis?*

The visible phenotype, in tandem with specific patterns of gene expression, suggests a whole-embryo disruption of ionoregulation, and therefore a novel AOP. As in mammals, fluid accumulation in freshwater fish is a consequence of water retention. Given that marine embryos are hyposmolar to their environment, the only plausible way for edema to accumulate is a gain of electrolytes from the surrounding seawater. This is consistent with the observed reduction in the expression of genes that code for the transporters the keep salt out of the embryo. The gain of ions most likely is largest over the yolk sac due to surface area. This then leads to higher ion levels in the yolk sac sinus, and a shift of free water from other distal tissues, such as the dorsal subdermal space. We have rephrased parts of the text in this section. E.g.: “Our findings suggest that crude old may disrupt MRC function…”

*Subsection: "A novel adverse outcome pathway: disruption of cholesterol homeostasis" – this subheading, and the following paragraphs, claim novelty, and appear to claim precedent for this discovery. Has oil exposure impacts on yolk utilization not previously been reported in the literature? I can find a publication from the 1990s after just a couple minutes searching. Furthermore, the studies presented here do not provide direct evidence for impacts on yolk mobilization or cholesterol homeostasis. E.g., how do the authors know that the apparently increased yolk size in Figure 1 isn't just a result of fluid accumulation stemming from observed edema?*

As we clarify in this revised submission, edema accumulates well before there are detectable changes in the haddock yolk mass. Further, it seems unlikely that free water would move into the yolk platelets. Rather, edema accumulates in the yolk sac sinus, external to the yolk syncytial layer. We are aware of only one examination of yolk absorption, by our collaborators Mark Carls and Jeep Rice (published in 1999 on oil-exposed herring embryos). However, they did not make a connection to circulatory defects or lipid metabolism as we have in the present study. As for direct evidence of disrupted cholesterol homeostasis, we are not sure what is more direct than up-regulation of HMG-CoA-reductase. This is the rate-limiting enzyme for cholesterol synthesis, and its expression is extremely sensitive to sterol levels.

*In the same section: "Pathway analysis was also consistent with a significant effect on cholesterol homeostasi." – The associated figure is missing. Also, staring at Figure 3—figure supplement 1 I see no pathways that include any terms obviously related to cholesterol homeostasis. Nor for that matter does Figure 1—figure supplement 1 indicate any enrichment of pathways having to do with ionoregulation, or craniofacial development,*

We have addressed this through the new IPA analysis and associated tables.

*Subsection “Unaltered gene expression in relation to visibly normal organs: lateral line and liver.”: "Similarly, the related KEGG pathways that are inclusive of these genes were relatively unaffected by oil exposure at all time points" – but what does this really mean? KEGG pathways (Figure 3—figure supplement 1 and Figure 3—figure supplement 2), as far as I can tell, don't implicate many (any?) of the mechanisms that the authors are building their story on – e.g., ionoregulation, cholesterol homeostasis, craniofacial development. If this paragraph was intended to serve as a test or confirmation that their conclusions/assertions in preceding paragraphs have merit, then this is unconvincing – and problematic for supporting the assertions that the other functions (ionoregulation, cholesterol homeostasis, craniofacial development) ARE supported by pathway-level analysis.*

We have addressed this through the new IPA analysis and associated tables.

*Discussion section:*

*"We identified specific changes in the expression of key genes involved in the function or morphogenesis of individual tissues and organs with visible abnormalities. Given unaltered gene expression associated with apparently unaffected structures such as the liver, the DEGs in oil-exposed haddock indicate a disruption of specific developmental processes, as opposed to non-specific effects (e.g., general developmental delay)." – this appears to me as a gross overstatement. There was no organ-specific analysis of gene expression. The authors measured gene expression in whole animals. They therefore are not in a position to make assertions about expression in the heart or liver, though of course they are free to propose hypotheses.*

We respectfully disagree, as captured (in part) in the discussion about null hypotheses above. While sophisticated tools are available for model species (e.g., organ-specific GFP expression) to isolate specific cell populations (e.g., by flow cytometry), this is practically impossible for a non-model species. A key aim of our study was to determine whether organ-specific changes in gene expression are detectable and consistent with the development of abnormal structures (phenotypic anchors). At issue is whether changes in gene expression, as visualized by whole-mount in situ hybridization, parallel quantitative changes in transcript abundance. Many previous studies of individual genes have confirmed this. As the reviewer pointed out earlier, the challenge is one of signal-to-noise, or whether gene regulation specific to a very small organ can be detected against a background of more abundant transcripts from a much larger tissue such as skeletal muscle. We have already confirmed this in our assessment of gene expression in normally developing haddock (Sørhus et al., 2016; Dev. Biol. 411:301). As structures became visible in the embryo over time (somites, heart, eye, etc.), we measured corresponding changes in the abundance of RNAs that are known to be associated with those structures via in situ hybridization. In this paper we show that this is also the case for these same structures that develop abnormally in response to crude oil. Consider, for example, *bmp10*. In situ hybridization in the three major vertebrate models (mouse, chick and zebrafish) has clearly shown that this gene is expressed exclusively in the heart during early development. In zebrafish, *bmp10* is expressed in ~ 100 cells. Consistent with this, the read-count data for *bmp10* in unexposed controls is just above the detection limit. In mutants with malformed hearts, *bmp10* upregulation is always visible by in situ hybridization. Indeed, changes in both the patterns and levels of gene expression – as assessed by in situs – are a fundamental tool in developmental genetics to relate molecular pathways to formation of structure. In situs are regularly used semi-quantitatively, and we predicted similar changes in the RNA-Seq data. We have addressed this concern with additional text in the Discussion section and a new table ([Supplementary-material SD1-data]) relating raw read-count data to tissue-specific expression.

*"Our data demonstrate a transcriptional cascade that is tightly linked to these defects in cardiac function (cardiomyocyte intracellular calcium cycling) and form (heart chamber growth)." – there are 4 genes that are slightly down-regulated at one timepoint preceding the emergence of altered cardiac phenotypes (6 dpf), and none deferentially expressed immediately post-organogenesis (10 dpf) when the proposed E-C uncoupling should be apparent. Does this represent the discovery/demonstration of a "transcriptional cascade"?*

This statement is not entirely accurate. The first change observed was a 4-fold upregulation of *bmp10*, an extremely potent morphogen. Subsequently we observed upregulation of the *bmp10* target gene *nkx2.5*. Defects in E-C coupling at the protein level are likely occurring as soon as PAHs accumulate in the embryo. We are proposing that E-C coupling and excitation-transcription coupling are simultaneously disrupted by PAHs, by processes that involve SR and nuclear calcium handling. The *bmp10*-initiated transcriptional cascade is the link between SR calcium cycling and morphogenesis.

*"Crude oil likely disrupts normal MRC function" – what is the evidence for this?*

*"MRC function could be impaired by a high metabolic cost of PAH degradation." – why? Evidence or rationale to support this?*

The evidence for this is the differential regulation of MRC genes prior to visible defects in circulation. We are suggesting that the high metabolic demand of osmoregulation (the reason MRCs are mitochondria-rich) is compromised by an abnormal metabolic demand for PAH metabolism. We (and others) have shown that PAH exposure leads to massive upregulation of cyp1a throughout the epidermis in the embryos of multiple fish species, including haddock. Thus the epidermis is the major proximal target organ for PAHs, with an associated metabolic cost. The cyp1a response is uniform throughout the epidermis where the MRCs reside. Still unresolved is whether the ATP demands for PAH metabolism competes with the ATP available for ion transport, or whether PAHs or metabolites (or other compounds) directly interfere with channels, transporters or pumps as they do in EC coupling.

*"In haddock embryos, fluid moves from the dorsal subdermal space to the yolk sac. At the larval stage, the permeable yolk sac membrane is replaced by the more resistant peritoneal cavity and body wall, causing fluid to move into the dorsal finfold and adjacent tissues." – I assume that the authors mean to write this as a proposal or hypothesis that is consistent with their observations?*

This sentence has now been rephrased as a hypothesis that is consistent with our observations.

*Discussion section paragraph nine: much of this is highly speculative. There are plenty of transcriptomics studies of oil exposure during fish development, that include edema as an endpoint. If the authors want to make these assertions perhaps they should check those other studies for altered regulation of VEGF-C.*

*Discussion paragraph ten: This entire paragraph reads as excessive speculation*

In fact, there is not an abundance of transcriptomic studies for fish exposed embryonically to crude oil. To our knowledge, the first was just recently published (Xu et al., 2016, Environ. Sci. Technol. 50:7842) while this paper was under initial review, by a team we have previously collaborated with in the context of Deepwater Horizon impacts to mahi mahi in the Gulf of Mexico. There are more studies looking at the transcriptomes of fish larvae (vs. embryos), including several on cod, also by collaborators. A comparison to mahi mahi would be tenuous, as they are much less sensitive than haddock to crude oil exposure, with an edema accumulation phenotype that is mild by comparison (Edmunds et al., 2015, Sci. Rep. 5:17326). The genes discussed in this paragraph are exceptional in that they stand apart from the overall dataset, and an established literature (which we cite) is available to provide some context for interpreting these results. Our intention is not speculation, but rather a suggestion that these genes should be a focus for future studies.

*Discussion paragraph eleven: "but tissue localization during craniofacial development in embryonic fish is needed to confirm a role in this adverse outcome pathway" – look up Planchart & Mattingly 2010 TCDD upregulates FOXQ1 in zebrafish jaw primordium*

We are aware that TCDD does indeed upregulate *foxq1* in zebrafish jaw primordium, but jaw malformation in TCDD-exposed zebrafish is entirely secondary to TCDD-induced heart defects. This has been rigorously demonstrated by a genetic gain-of-function approach (Lanham et al., 2014, Toxicol. Sci. 141:141). In the case of oil-exposed haddock, the craniofacial malformations are uncoupled from cardiac defects/edema formation, as discussed in Sørhus et al., 2016 (Sci. Rep. 6:31058). While Planchart and Mattingly did not demonstrate AHR-dependence of *foxq1*, this gene was shown to be AHR-responsive in a rat liver progenitor cell line (Faust et al., 2013, Arch. Toxicol. 87:681). Thus we cannot rule in or out a role for *foxq1* in craniofacial malformations, or expression in the epidermis or liver with the rest of the AHR battery. This will likely need to be eventually resolved by in situ hybridization.

*Discussion section: "First, we used known spatial mRNA distributions in model species (primarily zebrafish) to more accurately phenotypically anchor the transcriptome data for crude oil-exposed haddock" – this sounds fancy but there is no description of this in the methods*

The requested description was added to the Methods. We simply utilized zfin.org to identify expression patterns that have been determined in previous studies by the zebrafish community. The search tool on the zfin.org landing page allows for detailed advanced search capabilities, either by gene, tissue, developmental stage, etc. Absent data in the Zfin database, we searched PubMed for papers showing in situ data in either chick or mouse.

*Reviewer #2:*

*[…] Major concerns:*

*1) Given that haddock is not a model system, better explanation of the developmental time windows will be critical for most readers to understanding the context of the embryonic and larval developmental stages. In zebrafish 50% epiboly occurs at 5.5 hpf while in haddock it occurs at 3 dpf.*

This is an excellent suggestion. In response, we have added information on the timing of normal haddock development. We also provide a new figure (Figure 2) that describes the developmental onset of each phenotype investigated.

*2) Authors need to add Alcian images of the new facial phenotypes not listed in their previous work, the upper jaw is not shown in previously nor is the basicranium.*

The basicranium was shown (labeled as trabeculae crania/ethmoid plate) in our previous paper. In cod and haddock, the upper jaw is not at a sufficient stage of chondrogenesis to be detected by Alcian blue until much later in the larval period (45 days post hatch for cod; von Herbring 2001, J. Fish Biol. 59:767).

*3) The authors need to be more explicit in their logic in moving from a broad overview of the transcriptome and to genes that directly impact the phenotypes listed when those genes are not the most highly responsive. Why discuss the broader transcriptional changes (subsection “General patterns of gene regulation in response to crude oil”)?*

The addition of results from IPA has significantly expanded the broad overview aspect of this study. This demonstrates that the pathways that contain our individual genes of interest are among the most highly responsive to crude oil exposure.

*4) Without the ability to create genetic chimeras or cell-type specific transgenics to directly test the tissue target of PAHs effect on facial development, the authors need to soften their stance that PAH disrupts muscle development thereby affecting skeletal development. Crump and Schilling have shown that in zebrafish edn1 is expressed in and required in NCCs. The authors cannot rule out a direct NCC-PAH impact based only on their data (Discussion section).*

This is a fair point, and although we believe this is the most parsimonious explanation, we have qualified our interpretation. We discount a role for NCCs primarily because the phenotype does not appear to be a patterning defect, as would be expected from a disruption of NCC regulators. For example, in zebrafish, a loss of *edn1* dorsalizes the ventral jaw structure, and ectopic/excess *edn1* ventralizes the dorsal jaw structure.

*5) What do the authors mean by the "transcriptional response to disrupted E-C coupling was not a simple pattern of compensation" (subsection “Genes associated with defects in cardiac function and morphogenesis”)?*

See response to this same question from reviewer 1.

*6) Several the figures need to be reviewed. Two figure supplements are missing from the manuscript, though they are described in the text (subsection “A novel adverse outcome pathway: disruption of cholesterol homeostasis”). Figure 6 shows that myh1 is down regulated at 0 dph not 11 dpf as described in the text (subsection “Genes associated with craniofacial abnormalities”). The supporting datasets need to be organized with the supplemental Table 1 for readability. It is difficult to relate these datasets with the table in the current form.*

These figures were mislabeled in the previous version of the manuscript. Both text and figure labeling have now been corrected. Our intent was to show which 10 genes are most up-regulated and down-regulated at each several stages. To visualize when some genes were highly expressed at several stages, we collapsed the row were most up or down regulated at several stages during development. We have rewritten the Table legend to clarify this.

*Reviewer #3:*

*[…] Substantive concerns:*

*1) In the title, Abstract, and manuscript, the authors overreach when they invoke "chemical genetics" and "adverse outcome pathways" (AOPs). Chemical genetics involves high-throughput screening of libraries of individual compounds, a much more precise approach than the exposure to a complex chemical mixture performed here. Although nowhere defined by the authors, an AOP describes the entire sequence of events from a molecular initiating event, across multiple levels of biological organization, to an adverse outcome; it is synonymous with "mechanism of action" (see e.g. Ankley 2010 Envir. Tox. Chem. and Villeneuve 2011 Envir. Tox. Chem.). When the authors refer to AOPs they are actually referring only to adverse outcomes, not the pathways. Their gene expression data may help to inform our understanding of some AOPs associated with oil exposure, but they certainly have not "revealed novel AOPs." And contrary to the claim in the Abstract, it is not clear that they have "identified initiating events"-the specific chemical-protein interactions that lead to the gene expression and phenotypic changes that they report.*

Our view of chemical genetics is broader and not necessarily confined to high-throughput screens of small molecule libraries. This is what the field has evolved into, but chemical genetics is simply the premise that small molecules can produce abnormal developmental phenotypes in a manner similar to single-gene mutations. Indeed, the logic of the small molecule screening approach was derived from earlier focal studies of teratogens that caused malformations through specific interactions with developmental signaling molecules, such as cyclopamine-hedgehog signaling (Incardona et al., 1998, Development 125:3553). This history is captured in the Introduction to the first published zebrafish small molecule screen (Peterson et al., 2000, Proc. Nat. Acad. Sci. 97:12965). In traditional forward developmental genetics, molecular pathways are identified by relating mutant phenotypes to the physical location of the mutation on the chromosome. On one hand, chemical genetics is simply the use of compounds to alter gene function at the protein end, thereby working towards the same goal (pathway identification) in the opposite direction. Chemically induced phenotypes can provide a greater level of control over the timing of gene function, for example, and sidestep experimental challenges such as embryonic lethality. The power of genetics – chemical or otherwise – is derived from the anchoring of gene expression to a careful phenotypic quantification and anchoring to gene expression. This was the goal of the present study, and we believe the use of the term is defensible. We also wanted to emphasize the feasibility of this approach in real-world organisms to study traits that may be absent from the lab models typically used in high throughput screening. Finally, as a study focusing on gene-environment interactions, we draw additional comparisons to analyses of natural compounds. Cyclopamine was initially identified from a similarly complex mixture of compounds produced by a range plant (*Veratrum viridae*) that induced malformations in pregnant sheep.

We agree that a goal in AOP development is to describe a series of related events across biological scales. But this is a process that plays out across many studies conducted over years or decades of research. In this regard, the framework for scaling crude oil-driven adverse health impacts to fish is relatively advanced, as a consequence of > 25 years of science initially spurred by the 1989 Exxon Valdez spill in Prince William Sound. The early life stage-based cardiotoxicity AOP was the focus of a recent review (Incardona and Scholz, 2016, Aquat. Toxicol. 177:555). In the current paper, we refer to population-scale adverse outcomes as the motivation for conducting this research, to inform future fisheries management by the National Marine Fisheries Service in the U.S. and the Institute for Marine Research in Norway. In terms of the actual data generated, our current study clarifies the relationships between initiating events and organ- or organism-level adverse outcomes. In certain cases our findings represent clear initiating events or key events strongly supported by the literature – e.g., *bmp10* upregulation, consequent *nkx2.5* upregulation, and ventricular malformation. In other cases we propose steps in a possible AOP that will require additional study – e.g., water shifts in the developing neural tube, aquaporin up- or downregulation, and subsequent abnormal CNS development. Although semantic, the distinctions are important, and we have provided additional background on AOPs and their development in the revised text.

*2) The authors' approach, which is stated explicitly (Results section), is to interpret changes in haddock gene expression in relation to known zebrafish mutants, i.e. they focused only on specific genes known to be involved in development of the tissues affected by oil. While this is valuable, is there a more objective approach that might be used to identify unexpected associations between changes in gene expression and specific phenotypes? It seems as though they have not taken full advantage of the unbiased RNA-seq dataset in this regard.*

We have used the unbiased RNA-seq dataset as a tool. As described above (response to Editor), we conducted an extensive manual curation of all DEGs (Kmeans, pathways analysis, inspecting manually the genes underlying the most regulated pathways, most regulated genes) before focusing on a final set of genes. Moreover, we have since added the results from an unbiased IPA. The reviewer may be more concerned about false negatives, or missing unidentified genes that might play key roles in heart development, for example. Given current state of technology, this is likely an unavoidable limitation of doing functional genomics in a non-model species where it is extremely difficult to manipulate the embryo with the types of tools available to zebrafish. In addition, IPA will only make associations based on previously published data. Nevertheless, given strong positive results with known potent signaling molecules (*bmp10, wnt9b, edn1*) and key transcription factors, it seems unlikely that we are missing novel key genes. These genes alone are sufficient to explain the phenotypes. In future studies we intend to develop in situ hybridization probes for some of the highly regulated genes for which there are no tissue localization data available, to further define their role in the crude oil injury phenotypes.

*3) The authors over-interpret the connections between gene expression patterns and phenotypes, claiming cause-effect relationships in haddock from what are only associations. For example, Discussion section paragraph two claims a "tight linkage" between gene expression and cardiac defects. Whether the gene expression changes are causal or are secondary to the phenotypic changes is not clear. The authors could strengthen their arguments by being more explicit about the temporal and concentration-dependent associations between gene expression patterns and phenotypes, e.g. by adding a measure of phenotypic progression to Figure 4–Figure 7.*

As discussed in the responses to reviewer 1, we have provided additional data showing how changes in gene expression precede the developmental emergence of each visible phenotype. We have also revised the text to support our conclusions of tight cause-effect linkages based on gene dosage studies from zebrafish. Small changes in the expression of potent morphogens such as bmp10 and master transcriptional regulators such *nkx2.5* produce large changes in developmental trajectory. For morphogens in particular, this is how they work in the process of specifying cell fate. For the most potent ligand in the bmp family, *bmp10*, too much or too little protein leads to defective heart development. The cumulative data from both mouse and zebrafish indicate that it would be impossible to form a normal heart if *bmp10* were inappropriately upregulated too early in embryonic development, as we observed here. Morphogen effects are exquisitely dose-dependent, such that even less than 2-fold changes can lead to different target gene activation.

*4) The experimental design is not described sufficiently. The methods (paragraph one) refer to a previous paper, but it is not clear whether these samples are from the same experiment described in that paper, or just used similar exposures. Indeed, the oil concentrations in that paper are expressed differently than in the current manuscript. Even if the methods are the same, this paper should be a description of the experimental design, including exposure conditions, numbers of replicates, numbers of pooled embryos in each replicate, etc.*

We have clarified the source of the samples (the same as for Sørhus et al., 2016, Sci. Rep. 6:31058). Oil concentrations have also been corrected to match the previous paper.

*[Editors' note: further revisions were requested prior to acceptance, as described below.]*

*Reviewer #1:*

*Details of the experimental design and analysis are still not adequate. The authors refer the reader to the Sorhus 2016b publication for all aspects of experimental design. The authors should provide in the current manuscript at least the basics of experimental design in the methods section, including the number of doses, the specific developmental stages examined within each dose, and the number of biological replicates (replicate pools) within each dose*stage treatment. Furthermore, I have read Sorhus 2016b and it is not obvious WHAT were the exact contrasts without much effort on the part of the reader. The authors should make it easy to understand the basic of the experimental design.*

*Also, it is still not clear to me what was the unit of replication. I asked for clarification on the nature of replication previously. The revised manuscript reads "cDNA library preparation and sequencing was performed by the Norwegian Sequencing Centre (NSC, Oslo, Norway) on one pool from three replicate tanks per stage for each dose (low, pulse and high) plus control using the Illumina TruSeq RNA Sample Preparation Kit".*

*A treatment is a dose*stage. Were there replicate pools of embryos assayed per treatment? As far as I can tell, six developmental stages were profiled at each of 3 doses and a control, resulting in 24 treatments. I find 24 "samples" sequenced in the SRA (https://www.ncbi.nlm.nih.gov/Traces/study/?acc=SRP060012). This would suggest one pool per treatment, unless there are multiple samples embedded within each SRA entry. If there is just one sequenced pool per dose*stage treatment, then I recommend rejection of the manuscript for lack of replication of experimental units. If I misunderstand this, and there is in fact replication of pools of embryos within each dose*stage treatment, then I recommend the following:*

The 24 samples the reviewer is referring to is the samples from Sørhus et.al 2016a DevBio. The samples from this study, however, should now be available in the SRA database under this accession ID: PRJNA328092.

Additional information has been added to the Methods section. To make it clear: A total of 132 samples were sequenced, and 126 should be included in SRA (one pulsed group tank appeared to be an outsider and was excluded from further analysis, a matter that is thoroughly explained in the Results section Sørhus 2016b). We have clarified this further in the text. " cDNA library preparation and sequencing was performed by the Norwegian Sequencing Centre (NSC, Oslo, Norway) using the Illumina TruSeq RNA Sample Preparation Kit.A total of 132 samples were sequenced and 126 were subjected for analysis: Three (control, low and high dose) or two (pulse) biological replicates for each dose from 6 stages during and after embryonic exposure and three biological replicates for each dose from 5 stages during larval exposure.”

*Subsection “Structure of pelagic larvae and visible phenotypes associated with crude oil exposure”: "with 96% of high dose animals showing abnormal phenotypes, ranging to ~ 60% for pulse dose and ~ 35% for the low dose."*

*The authors should consider preparing a visual that summarizes this distribution of phenotypes across doses. E.g., stacked bar chart (or pie chart), including proportion of animals showing each phenotype, with dose as series. And perhaps separate plots for different developmental stages.*

We understand the benefit of having a visual summary in this paper. However, the Sørhus et.al 2016b paper also includes both figures and tables showing a clear dose dependent effect for abnormal phenotypes, functional and development cardiac abnormalities and edema accumulation. Thus, including the exact same figures and tables should therefore not be necessary.

*Subsection “Oil-induced changes in gene expression during embryonic development.”: "p > 0.05" Is it not more standard to notate as p<0.05?*

Corrected

*Subsection “General patterns of gene expression in response to crude oil”: "At all stages, the subcategory of Organismal Development or Embryonic Development (henceforth combined as Development) was in the top 5 Diseases and Bio Functions category under Physiological System Development and Function with p values ranging from 10-3 to 10-19 " Are categories/subcategories of functions from IPA? Authors should state this.*

Yes, these are the classifications provided in the IPA output. We have provided additional detail in the Methods section to highlight this, in addition to stating this in the main text.

*In the same subsection: "Pathway enrichment was dose-dependent and clearly associated with the frequencies of abnormal phenotypes ([Supplementary-material SD1-data])." It would appear that reviewers do not have access to these supplementary files. This is frustrating.*

We have confirmed that these lists have been uploaded, and we have requested them to be available for the reviewers in our letter to editor.

*Subsection “Genes associated with defects in cardiac function and morphogenesis”: "overexpression of bmp10, nkx25, or tbx3 is associated with serious heart defects in other vertebrates." This needs a citation*

The citations Chen 2006, Ribeiro 2007 and Tu 2009 have been inserted.

*Discussion section: "Two major initiating events for crude oil-associated cardiac defects during fish development are chemical blockade of IK_r_ repolarizing potassium currents, (encoded by kcnh2) and disruption of intracellular calcium handling, the latter culminating in sarcoplasmic reticulum (SR) calcium depletion through effects on either RyR or SERCA2 (encoded by ryr2 and at2a2, respectively). In the fully formed heart, these pharmacologic effects impair cardiac function by inducing arrhythmia and reducing contractility." Citations are needed in this section.*

The citations Brette 2014 and Incardona 2009 and 2014 have been inserted.

*Subsection “Extraction of mRNA, RNA sequencing and bioinformatics”: "150 key genes involved in cardiac and craniofacial development and cardiac function were assembled." Please provide a table of these genes, and relevant citations describing their relationships to key phenotypes. The authors state in their rebuttal "The lists are provided in a new table, [Supplementary-material SD1-data]." It would appear that I unfortunately do not have access to review these files (or I'm somehow just looking in the wrong place?). If not already done, authors please make sure these files are detailed with the phenotype relationship and citations to all relevant literature (e.g., defend criteria for including a gene in your curated set).*

Hopefully the reviewers have now been allowed to view the supplementary files. The table include name and relation to phenotype. A thorough functional description of the genes including citations are provided in Sørhus et al. 2016a. We also adjusted the supplementary file legend to: [Supplementary-material SD1-data]: Manually curated list of genes involved in cardiac development and function and craniofacial and bone development and maintenance. For references see Sørhus et. al 2016a, DOI: 10.1016/j.ydbio.2016.02.012.

SP, swissprot; GB, genebank.

*In the same subsection: "The contigs" is this the reference cod sequence, or the haddock RNA-seq read sequences?*

This refers to the cod genes. We have now clarified this in the text: “The cod genes[…]”

*The authors state in their rebuttal "Although the effects of oil exposure on salt and water balance in fish embryos have not been examined," This is perhaps true, but it has been examined in adults, and this may be worth noting in the manuscript. E.g.: Kennedy CJ, Farrell AP (2005) Ion homeostasis and interrenal stress responses in juvenile Pacific herring, Clupea pallasi, exposed to the water-soluble fraction of crude oil. Journal of Experimental Marine Biology and Ecology 323, 43-56.*

Very good point. This point has now been added to the manuscript text.

*Regarding the author's rebuttal "As for direct evidence of disrupted cholesterol homeostasis, we are not sure what is more direct than up-regulation of HMG-CoA-reductase" Up regulation of a gene is not a direct measure of altered cholesterol homeostasis.*

Actually, in this case up-regulation of HMG-CoA reductase (hmdh) is unequivocally a direct measure of altered cholesterol homeostasis. This is so well established it can be found in any cell biology text book, and is the subject of the Brown and Goldstein review we cited in the Results section when first mentioning hmdh. However, we have made this more explicit in the relevant Discussion section with the statement “Cellular sterol levels are tightly controlled by membrane-bound transcription factors, which are cleaved and activated when membrane cholesterol levels fall, leading to transcription of hmdh, the rate-limiting enzyme for cholesterol synthesis”, citing Brown and Goldstein again. Like CYP1A induction in response to aromatic xenobiotics, this is a classic example of transcriptional control of metabolism, harking back to Jacob and Monod and the lac operon in E. coli.

*Reviewer #3:*

*I would quibble with a few of the responses to my original comments, but I don't view these issues as serious enough to derail publication of the paper. Nevertheless, I point them out for the authors' consideration.*

*1) Regarding what is an "initiating event" in an adverse outcome pathway (AOP), I disagree with the authors' claim (response to review 3) that bmp10 upregulation is such an initiating event. It is an early event, certainly. But the true initiating event is the chemical-protein interaction that causes bmp10 to be upregulated. That event has not been identified in this paper.*

That is true. We do not in this paper provide the direct evidence for the first initiating event. However, in the paper we suggest that the effect on ETC, resulting in an inappropriate calcium activation of myocardin as the initiating event, leading to the up-regulation of bmp10.

*2) With regard to my questioning of their statement claiming that it has not been clear that oil influences mRNA levels as part of a developmental phenotype (Discussion section): The authors pointed out that the several previous papers I noted as showing oil affecting mRNA expression in fish did not involve measurements made in embryos. However, they did not provide this explanation in the revised manuscript, despite the fact that two of the three reviewers raised the same question. In addition, in their response the authors mention a more recent paper, published (27 June) prior to the original submission of this manuscript (30 Aug), that does include transcriptomic analysis of fish embryos exposed to oil (Xu et al. ES&T). It is unfortunate that the authors did not take advantage of the opportunity to better explain their original statement or compare their results with the prior work in embryos.*

The paper by Xu et al. in ES&T only looked a gene expression in hatched larvae, not at earlier time point during embryogenesis, and in particular prior to the appearance of abnormal phenotypes. Therefore, none of the papers in question are really comparable. We are uncertain of the value in adding text and additional citations to point this out.

*3) The manuscript would be improved by a clearer summary of the experimental design so that the readers don't have to dig out the other paper. And the point about replication is important – this needs clarification.*

Additional information has been added to the Materials and methods section.

[Editors' note: further revisions were requested prior to acceptance, as described below.]

*Reviewer #1:*

*The revisions, in general are fine. However, I should note that I have been generally frustrated by the struggle with the authors to include all relevant information, accessibly, within this manuscript. Far too many papers are published these days that have incomplete description of methods. Also, many papers refer the reader to other papers to find out details of the methods. This results in un-necessary additional work for the reader – especially un-necessary since all journals these days have supplemental sections that allow authors to include all relevant information, without cluttering up the main manuscript. The authors are still insisting on sending the review on a hunt through their previous papers to find relevant information ("A thorough functional description of the genes including citations are provided in Sørhus et al. 2016a."). This is not too much to ask. We should all strive to make our research more transparent, and more reproducible.*

A reference column has been added to the [Supplementary-material SD1-data] including one or more reference for every gene.